# Steroid hormone signaling activates thermal nociception during *Drosophila* peripheral nervous system development

Jacob S Jaszczak[1,2], Laura DeVault[1,3], Lily Yeh Jan[1,2], Yuh Nung Jan[1,2]*

[1]Department of Physiology, Department of Biochemistry and Biophysics, University of California, San Francisco, San Francisco, United States; [2]Howard Hughes Medical Institute, Chevy Chase, United States; [3]Department of Developmental Biology, Washington University Medical School, Saint Louis, United States

**Abstract** Sensory neurons enable animals to detect environmental changes and avoid harm. An intriguing open question concerns how the various attributes of sensory neurons arise in development. *Drosophila melanogaster* larvae undergo a behavioral transition by robustly activating a thermal nociceptive escape behavior during the second half of larval development (third instar). The Class IV dendritic arborization (C4da) neurons are multimodal sensors which tile the body wall of *Drosophila* larvae and detect nociceptive temperature, light, and mechanical force. In contrast to the increase in nociceptive behavior in the third instar, we find that ultraviolet light-induced $Ca^{2+}$ activity in C4da neurons decreases during the same period of larval development. Loss of ecdysone receptor has previously been shown to reduce nociception in third instar larvae. We find that ligand-dependent activation of ecdysone signaling is sufficient to promote nociceptive responses in second instar larvae and suppress expression of *subdued* (encoding a TMEM16 channel). Reduction of *subdued* expression in second instar C4da neurons not only increases thermal nociception but also decreases the response to ultraviolet light. Thus, steroid hormone signaling suppresses *subdued* expression to facilitate the sensory switch of C4da neurons. This regulation of a developmental sensory switch through steroid hormone regulation of channel expression raises the possibility that ion channel homeostasis is a key target for tuning the development of sensory modalities.

*For correspondence:
YuhNung.Jan@ucsf.edu

**Competing interest:** The authors declare that no competing interests exist.

## Editor's evaluation

The authors describe in vivo analyses of an intriguing steroid-mediated development shift in the sensitivity of *Drosophila* larvae to noxious stimulation as they move from the L2 to the L3 instar stage. Experiments and observations presented show that the steroid hormone ecdysone regulates nociceptor activity in the peripheral nervous system by suppressing expression of a gene named subdued, which encodes a membrane protein of the TMEM16 channel family.

## Introduction

Detection of internal and external stimuli depends on developmental programing of the proper physiological and morphological properties of sensory neurons. It is an intriguing open question as to how the characteristics of sensory transduction arise during development, and which neuronal properties are crucial for the initiation of sensory detection.

Nociception, the ability to detect and escape potentially harmful stimuli, is a highly conserved function of sensory systems present in all animals (*Arenas et al., 2017*). A nociception system is composed of nociceptors, which are neurons with molecular components that directly sense harmful stimuli; a

**eLife digest** During their lives, animals encounter a broad range of stimuli from their surroundings including heat, light and touch. The ability to appropriately respond to such stimuli is crucial for survival as it allows the animals to avoid predators and other dangers, locate food and shelter, and find mates.

Fruit fly larvae are a useful model for studying how animals respond to unpleasant (known as painful) heat stimuli. When something hot touches a larva, the larva rolls away to avoid the stimulus. The heat stimulates electrical activity in a type of neuron known as C4da neurons on the surface of the larva. Ultraviolet light and several other stimuli are also able to activate electrical activity in C4da neurons, resulting in the larvae changing the direction they move to avoid the stimuli.

Only older fly larvae respond to painful heat stimuli and previous studies found that a hormone receptor protein is required for this response. However, it remains unclear how this response develops as the larvae age.

Jaszczak et al. studied the behavior of fly larvae and electrical activities of C4da neurons in response to painful heat and ultraviolet light. The experiments found that painful heat triggered more rolling behavior from older larvae than those of younger larvae. In contrast, ultraviolet light triggered lower levels of electrical activity in the C4da neurons of older larvae than those of younger larvae. The team raised the levels of a hormone known as ecdysone and found that this increased the rolling behavior in younger larvae. They then increased the amount of receptor protein for this hormone in the neurons and found that it decreased the levels of another protein called Subdued in the C4da neurons. This in turn increased the neurons' response to painful heat and decreased their response to ultraviolet light.

Jaszczak et al. propose that as the larva develops, ecdysone reduces the levels of Subdued, which promotes C4da neurons to switch their sensitivity from detecting ultraviolet light to painful heat. In the future, better understanding of what causes pain sensations in developing animals will help us search for factors that cause long-term pain conditions in humans.

processing center where the stimuli are interpreted, typically the neural circuits of the central nervous system (CNS); and the reflexive neuromuscular circuit which generates a reaction to escape injury (*Baliki and Apkarian, 2015*). Development of sensory systems requires the coordination of processes on the cellular, morphogenic, and circuit level.

Thermal nociception generates an avoidance behavior, such as a fast limb retraction or change of movement trajectory. Previous studies of thermal nociception behaviors and neural activity demonstrate a change in nociceptive responses during development of both mice and *Drosophila*. For example, a mouse paw exhibits low sensitivity to painful heat early in postnatal development, but the rate of reflex gets faster as the animal develops (*Jankowski et al., 2014*; *Ford et al., 2019*). A developmental transition is also observed in nociceptor activity, as revealed by the change in the number of heat-sensitive C-fibers with age in mice (*Hiura et al., 1992*). Likewise, a behavioral transition in heat thermal nociception has been observed in *Drosophila melanogaster* larvae (*Sulkowski et al., 2011*). *Drosophila* third instar larvae exhibit a stereotyped 'corkscrew' behavior, when a thermal probe heated above 38°C touches the cuticle (*Tracey et al., 2003*; *Babcock et al., 2009*). However, this behavior is not observed in second instar or earlier larval stages, and appears to be acquired only during the last stage of larval development.

The larval nociceptive behavior has been a useful model for dissection of the molecular mechanisms and circuitry of nociception (*Himmel et al., 2017*). Among the heat-sensitive neurons in the larval cuticle (*Liu et al., 2003*) are the Class IV dendritic arborization (C4da) neurons, which function as the primary nociceptors for noxious heat-induced escape behavior (*Tracey et al., 2003*; *Hwang et al., 2007*). C4da neurons are multimodal sensors for harmful stimuli, with the capacity to respond to multiple forms of sensory information. In addition to responding to temperature, C4da neurons respond to high-intensity blue light (*Xiang et al., 2010*), noxious UV light (*Xiang et al., 2010*; *Gu et al., 2019*), and harsh touch (*Zhong et al., 2010*).

While noxious heat and touch both elicit the rolling escape behavior, stimulation of C4da neurons with short wavelength (blue or UV) light causes a locomotion trajectory avoidance behavior. These different stimuli cause different electrical signals in C4da neurons. Noxious heat causes high-frequency

spike trains interspersed with quiescent periods, in contrast to blue light which elicits lower frequency firing with few quiescent periods in third instar larvae (*Terada et al., 2016*). Therefore, distinct behaviors appear to be encoded by the C4da neurons according to their differential responses to differing stimuli.

The majority of studies of nociception in *Drosophila* larvae have focused on the last (i.e., the third) larval instar, where nearly all larvae will perform the behavioral response. However, Sulkowski et al. found that during the first half of larval development (i.e., the first and second instars), larvae exhibited either no or a low rate of thermal nociceptive behavior, leading to the hypothesis that a developmental transition occurs between the second and third larval instars (*Sulkowski et al., 2011*). The mechanisms which regulate this transition remain unknown. Optogenetic activation of C4da neurons can induce the corkscrew behavior or a locomotion trajectory avoidance behavior in third instar larvae. However, optogenetic activation could only induce a locomotion trajectory avoidance behavior in the second instar. This difference between second and third instars in the optogenetic activation of behavior, taken together with the difference in coding activity in C4da neurons in response to thermal stimulation or light (*Terada et al., 2016*), raises the possibility that the developmental transition may involve a change in sensory coding. Supporting this possibility, misexpression of low temperature activated TrpA1 (isoform A) channels in the C4da neurons enables second instar larvae to exhibit corkscrew behavior in response to an innocuous temperature stimulus (*Luo et al., 2017*). It is conceivable that the sensory switch may be caused by a cell autonomous change in the C4da neurons. Whether C4da neurons change in their neural activity during the developmental transition from second to third instar is unknown, and the regulators of such transition are unknown as well.

Transcription factors facilitate the formation of distinct neuronal functions (*Parrish et al., 2014*) and regulate the dynamic expression profiles of many developmental transitions between and during larval instars (*Sullivan and Thummel, 2003*). One such transcription factor is the nuclear hormone receptor ecdysone receptor (EcR). EcR forms a heterodimer with ultraspiracle (USP) and binds the ligand ecdysone, a steroid hormone, to coordinate a wide variety of developmental programs (*Yamanaka et al., 2013*). EcR and USP are required for thermal nociception in the third instar (*McParland et al., 2015*), but the mechanism and role of ecdysone signaling in the second instar have not been determined.

EcR expression in C4da neurons has been reported at the beginning and end of larval development (*Kuo et al., 2005*; *Ou et al., 2008*; *Kirilly et al., 2009*; *McParland et al., 2015*). In the absence of ligand, EcR and USP distribute between the cytoplasm and nucleus, while ligand binding increases nuclear localization (*Nieva et al., 2007*; *Nieva et al., 2008*). Without ligand binding, EcR and USP recruit nucleosome remodeling proteins to compact the chromatin and locally suppress transcription. When bound to ecdysone, EcR-USP recruit coactivators, such as histone acetylases, to expand the chromatin structure and promote transcription. In this way, steroid hormone signaling can create positive and negative feedback to promote cascades of transcriptional programs at developmentally precise periods (*Sullivan and Thummel, 2003*; *Hill et al., 2013*). Whether the ligand-induced activity and transcriptional regulation are involved in EcR-mediated thermal nociception remains to be determined.

In this study, we performed a series of experiments to determine the activity of the C4da neurons during the developmental nociceptive transition as well as the mechanisms which regulate this sensory switch. These experiments demonstrate that steroid hormone signaling in nociceptive neurons regulates the channel composition in C4da neurons to modulate the sensory properties and behavioral response at distinct stages of larval development.

## Results

### C4da neuron sensory switch between second and third larval instars is modality specific

C4da neurons are multimodal nociceptive sensors of thermal, short wavelength light, and mechanical stimuli (*Himmel et al., 2017*). While first instar and third instar larvae respond at similar frequency to mechanonociceptive stimuli (*Almeida-Carvalho et al., 2017*), larvae do not robustly activate the thermally triggered nociceptive behavior until the last (third) instar of development (*Sulkowski et al., 2011*). Whether the C4da neurons respond to short wavelength light changes during larval

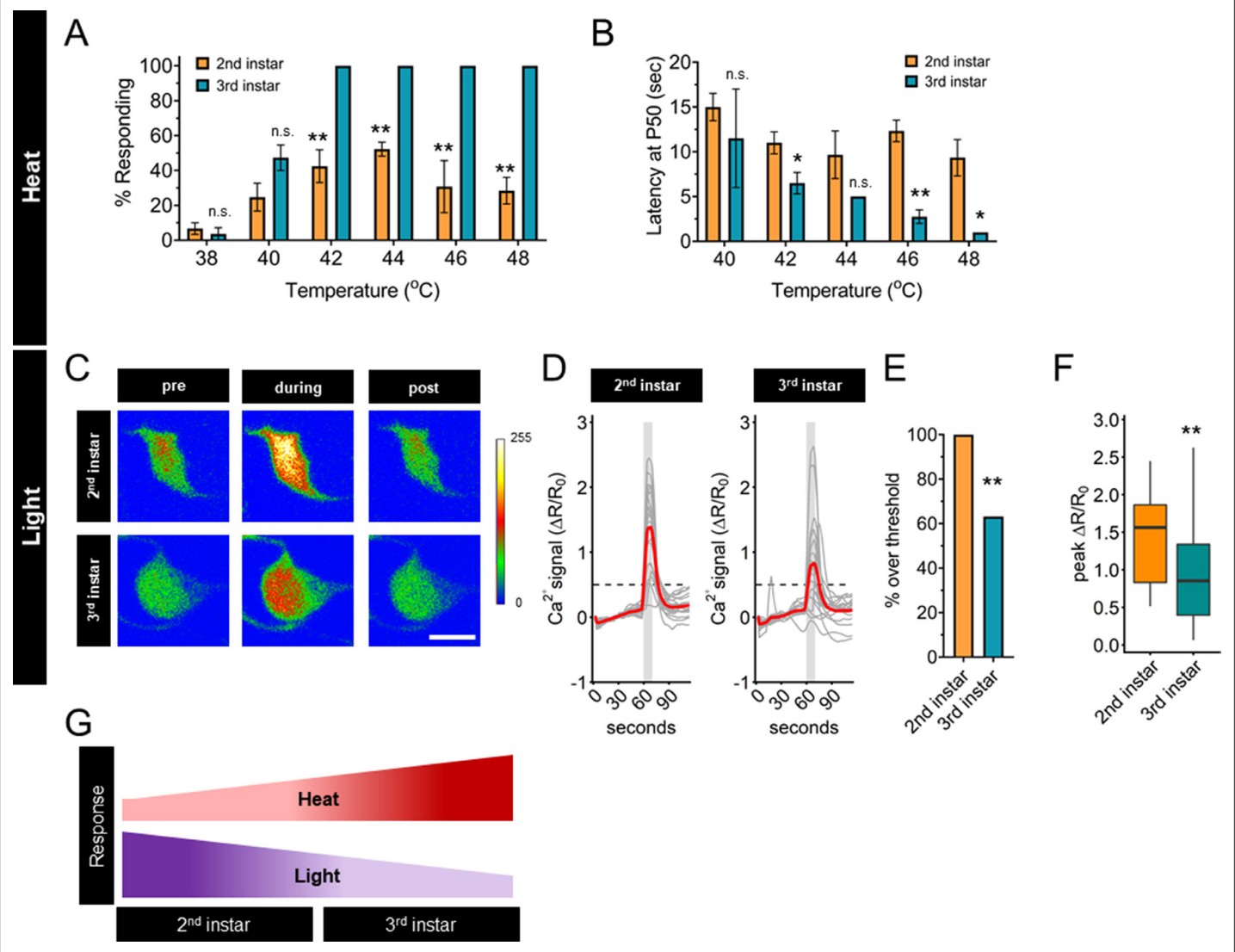

**Figure 1.** Thermal nociception behaviors increase during larval development, while UV-A light response decreases. (**A**) Groups of age-matched control larvae were touched with a thermal probe at different temperatures. The nociceptive response was scored if a 360° roll occurred during 20 s of stimulus. (**B**) Latency by which 50% of the responding population of larvae has had a nociceptive behavior response. (**C**) Representative GCaMP6s images of second and third instars in soma Class IV dendritic arborization (C4da) neurons. Scale bar = 10 µm. Periods of UV-A treatment program: Pre = before UV-A stimulus. During = period of UV-A light stimulus. Post = after the end of UV-A stimulus. (**D**) Individual traces (gray lines) and means (red lines) of $Ca^{2+}$ activity calculated by ratiometric change from baseline. Gray column indicates period of UV-A stimulus. Dotted line indicates threshold level for % over threshold calculations. (**E**) Percent of soma with $Ca^{2+}$ activity over the threshold level. (**F**) Peak $Ca^{2+}$ activity during the periods of UV-A treatment programs. (**G**) Heat and light development have opposite trends during larval development. (**A**) $n$ = 2–4 staging replicates of 15–20 larvae were tested for each age and temperature. (**C–F**) Second instar $n$ = 21 neurons. Third instar $n$ = 19 neurons. (**A, B**) Student's $t$-test. (**E**) Fisher's exact test, one-sided. (**F**) Student's $t$-test. *$p < 0.05$, **$p < 0.01$.

The online version of this article includes the following figure supplement(s) for figure 1:

**Figure supplement 1.** Thermal nociceptive sensitivity increases across temperatures from second to third instar.

development is unknown. To investigate whether this developmental transition is specific to thermal nociception, we sought to compare the responses of second and third instar larvae to thermal and light stimuli.

First, we measured the frequency and latency of the nociceptive behavior in second and third instar larvae across a range of nociceptive temperatures. Compared to third instar larvae, second instar larvae had a lower response frequency of nociceptive behavior across all nociceptive temperatures above 40°C (*Figure 1A*). Among the larvae which did display a nociceptive behavior, second

instar larvae had a longer latency before initiating the nociceptive response than third instar larvae (*Figure 1B*, *Figure 1—figure supplement 1A, B*). Thus, second instar larvae displayed reduced thermal nociceptive behavior across a wide range of temperatures.

High-intensity UV light exposure of third instar larvae causes behavioral responses, increased $Ca^{2+}$ activity, and patterns of spike trains that are different from those caused by nociceptive temperatures (*Xiang et al., 2010*; *Terada et al., 2016*). The small size of second instar larvae makes electrophysiological recording of C4da neurons prohibitive, therefore, we sought to compare the magnitude of the UV-induced $Ca^{2+}$ response. To determine if the UV light-induced $Ca^{2+}$ activity response of C4da neurons changes during development, we measured GCaMP6s signals of these sensory neurons in second and third instar larvae in response to 405 nm (UV-A) light stimulation. In contrast to the transition of thermal nociceptive behavior, UV-A stimuli-induced $Ca^{2+}$ activity of C4da neurons in second instar larvae was higher than that in third instar larvae (*Figure 1C, D*). This pattern, of third instar C4da neurons having lower UV-A-induced $Ca^{2+}$ activity than second instar C4da neurons, was validated by quantifying both the number of neurons which surpassed a threshold value of UV-A-induced $Ca^{2+}$ activity and the magnitude of the peak $Ca^{2+}$ activity (*Figure 1E, F*). Thus, the UV-A-induced $Ca^{2+}$ activity decreased with larval development, in contrast to the increase of thermal nociceptive neuronal response from second to third instar (*Figure 1G*). Based on these observations, we conclude that regulation of the sensory switch during the second and third instar development is modality specific.

## Steroid hormone ecdysone promotes the development of thermal nociception

Having found modality-specific sensory responses during the second to third instar developmental transition, we sought to find regulators of the sensory switch in C4da neurons by first examining the thermal nociceptive transition. The steroid hormone ecdysone regulates multiple developmental events during the second and third instars, with peaks in ecdysone titer triggering molting events and developmental checkpoints (*Rewitz et al., 2013*). Ecdysone is the ligand for the EcR with three isoforms (A, B1, and B2). These isoforms are identical in the DNA- and ligand-binding domains but differ in the activation function 1 (AF1) domain (*Figure 2—figure supplement 1A*). C4da neurons express isoforms EcR-A and EcR-B1 at embryonic stages, during the third instar, and dynamically during pupal development (*Kuo et al., 2005*; *Ou et al., 2008*; *Kirilly et al., 2009*; *McParland et al., 2015*). EcR-B1 regulates dendrite growth (*Ou et al., 2008*) and pruning (*Kuo et al., 2005*; *Kirilly et al., 2009*). EcR-A is required for thermal nociception in the third instar, as revealed by the greater reduction of nociception caused by RNAi knockdown of EcR-A as compared to RNAi knockdown of EcR-B1 (*McParland et al., 2015*). Therefore, we investigated the role of ecdysone and EcR-A in the developmental transition of thermal nociceptive behavior.

Nuclear localization of EcR increases during the end of larval development; localization is greater in late third instar larvae and pupae as compared to early third instar larvae (*Kuo et al., 2005*; *Kirilly et al., 2009*). However, the expression and localization of EcR isoforms during the second larval instars are unknown. To investigate EcR dynamics during the developmental period with different levels of thermal nociception, we examined the localization of EcR in second and third instar larval C4da neurons with an antibody which recognizes a common domain of all EcR isoforms. Consistent with previous findings (*Kuo et al., 2005*; *Ou et al., 2008*; *Kirilly et al., 2009*; *McParland et al., 2015*), we observed nuclear localization of EcR in wandering third instar larvae (*Figure 2A, B*). Earlier in larval development, EcR was evenly distributed between the nucleus and cytoplasm in the second instar (2 and 8 hr After L2 Ecdysis) and at the beginning of the third instar, 2 hr After L3 Ecdysis (AL3E). Nuclear localization increased 8 hr AL3E and in wandering third instar larvae (*Figure 2A, B*). Similar EcR localization dynamics could be detected for specific EcR isoforms, as revealed by antibodies specific to EcR-A (*Figure 2C, D*) and EcR-B1 (*Figure 2E, F*). Quantification of total EcR immunoreactivity in C4da neurons did not reveal any significant change during the second and third instar development (*Figure 2—figure supplement 1B*). These observations suggest that while EcR is present in C4da neurons throughout the second and third instars, the nuclear localization begins to increase early in the third instar.

Changes in ecdysone synthesis and systemic ecdysone titer orchestrate critical periods of *Drosophila* development. Pulses of ecdysone in the first and second larval instars increase systemic titers and promote expression of genes required for molting, while a series of three pulses of ecdysone in the

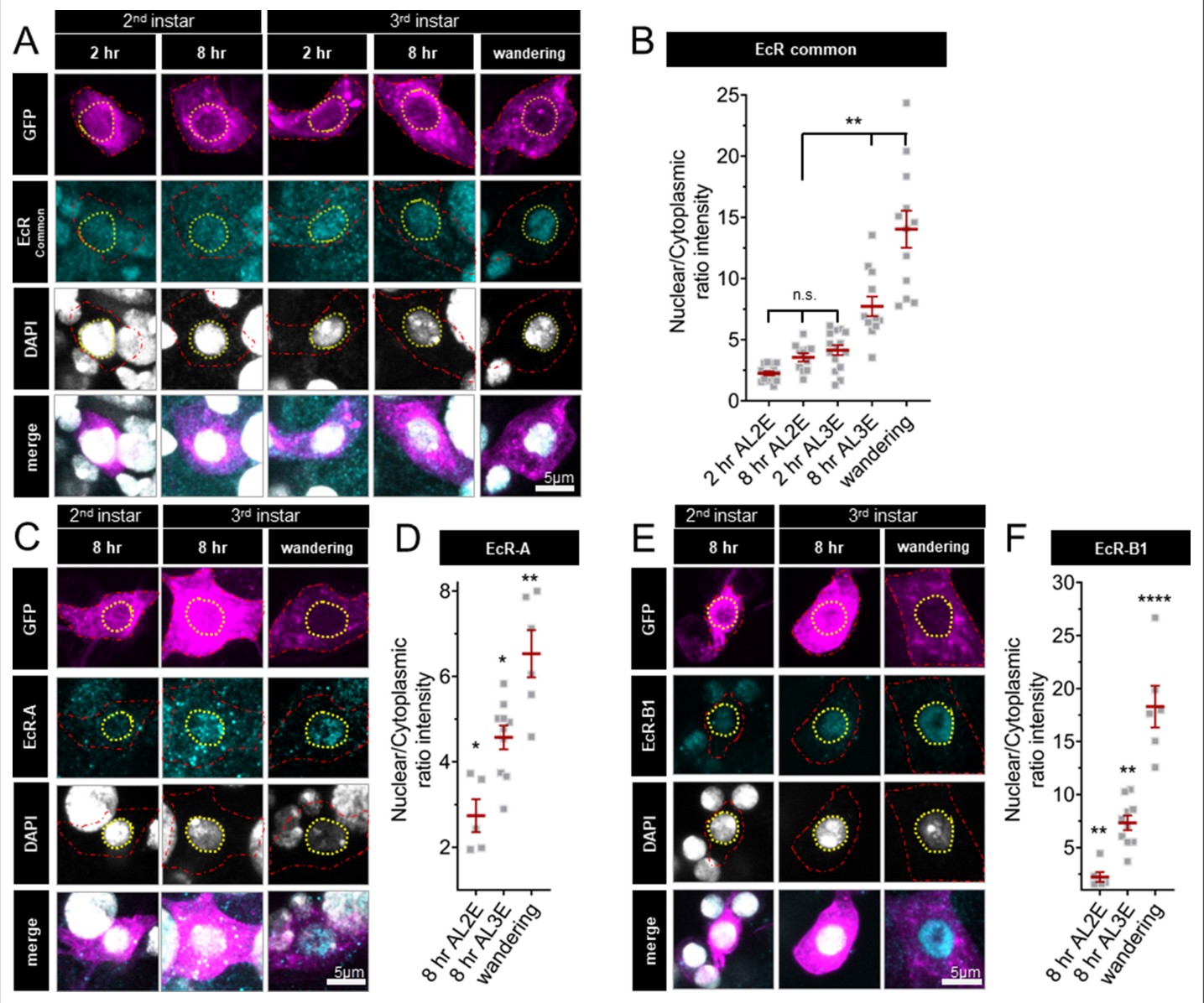

**Figure 2.** Ecdysone receptor (EcR) nuclear localization in Class IV dendritic arborization (C4da) neurons increases early during the third instar. (**A**) EcR immunohistochemistry with EcR-common antibody (**C**) EcR-A antibody, and (**E**) EcR-B1 antibody. Red dashed line outlines C4da neurons and yellow dashed line indicates position of nucleus. C4da neurons are labeled by ppktd-GFP. EcR nuclear localization quantified by nuclear to cytoplasmic ratio of intensity with (**B**) EcR-common and (**D**) EcR-A, and (**E**) EcR-B1 antibody. After L2 Ecdysis (AL2E), After L3 Ecdysis (AL3E). One-way analysis of variance (ANOVA) with Bonferroni post test. *p < 0.05, **p < 0.01, ****p < 0.0001.

The online version of this article includes the following figure supplement(s) for figure 2:

**Figure supplement 1.** Domains of ecdysone receptor (EcR) isoforms and total EcR in Class IV dendritic arborization (C4da) neuron early during the second and third instars.

third instar control developmental checkpoints and developmental preparation for metamorphosis (***Warren et al., 2006***; ***Kannangara et al., 2021***). To determine whether the increased thermal nociception in third instar larvae is driven by systemic ecdysone, we tested whether increasing steroid hormone titer during the second instar is sufficient to cause precocious development of thermal nociceptive behavior. Feeding larvae food supplemented with the metabolically active 20-hydroxyecdysone (20E) increases ecdysone signaling in larvae (***Colombani et al., 2005***). Thus, we transferred larvae to 20E supplemented food and measured larval thermal nociceptive behavior 8 hr later, while the larvae were still in the second instar. Larvae fed with 250 μg/ml 20E had a higher rate of behavioral response to

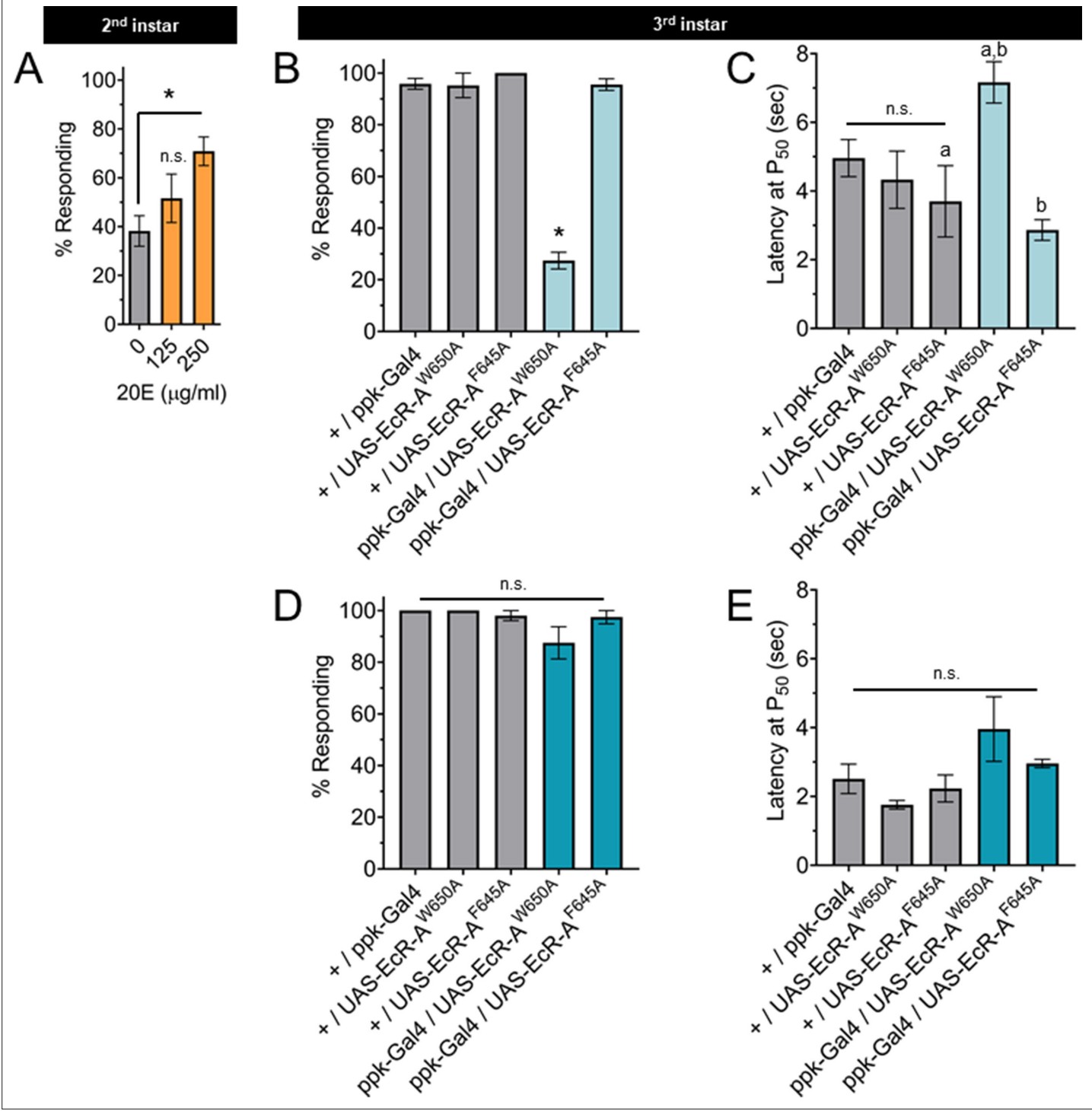

**Figure 3.** The nociceptive transition is ecdysone ligand activity dependent. (**A**) Percent of second instar larvae population displaying nociceptive behavior when fed 20-hydroxyecdysone (20E). (**B, D**) Percent of population displaying nociceptive behavior when overexpressing ligand-binding mutant ecdysone receptor (EcR)-A (ppk-Gal4/UAS-EcR-A -W650A), or coactivator mutant EcR-A (ppk-Gal4/UAS-EcR-A-F645A). Third instar larvae with (**B**) 42°C nociceptive probe or (**D**) 46°C nociceptive probe. (**C, E**) Latency by which 50% of the responding population displaying nociceptive behavior when overexpressing mutant EcR-A. Third instar larvae with (**C**) 42°C nociceptive probe or (**E**) 46°C nociceptive probe. One-way analysis of variance (ANOVA) with Tukey's post test. *, a, or b = p < 0.05. (**A**) $n > 35$ larvae for each treatment. (**B**) $n$ = 2–4 staging replicates of 15–20 larvae were tested for each genotype.

thermal nociceptive stimuli as compared to larvae fed with 125 μg/ml 20E or vehicle alone (*Figure 3A*), indicating that increasing ecdysone titer in the second instar is sufficient to promote the development of thermal nociception.

The binding of ecdysone to EcR causes the recruitment of coactivators and the loss of corepressors. In the third instar, reduction of EcR-A expression reduces thermal nociception (*McParland et al., 2015*). Therefore, we tested the effects of disrupting the EcR-A domains responsible for either ligand binding or coactivator recruitment on nociception, by characterizing mutants with alanine substitution in either the ligand-binding domain (EcR-A-W650A) or the coactivator recruitment domain (EcR-A-F645A) (*Cherbas et al., 2003*). The W650A mutation prevents ligand binding and disrupts both derepression and activation. The F645A mutation cannot mediate activation, but retains the ligand-binding capacity (*Hu et al., 2003*). Therefore, differences between the effect of overexpression of these mutant EcR-A constructs are likely due to differences in ligand-binding ability. Mutant forms of EcR can function in a dominant negative capacity by outcompeting the endogenous wild-type EcR, thereby inactivating the endogenous receptor function (*Cherbas et al., 2003*). In this way, overexpression of these mutant EcR-A constructs can distinguish the requirements of coactivation (disrupted by F645A) or ligand-induced derepression (disrupted by W650A). When assaying for a dominant negative mutant phenotype, we observed that expression of EcR-A-F645A in C4da neurons did not change nociceptive behavior in third instar larvae, while expression of EcR-A-W650A reduced the number of larvae exhibiting nociceptive behavior in third instar larvae (*Figure 3B*). Of the larvae that did exhibit nociceptive behavior, expression of EcR-A-W650A did not alter their response latency relative to the W650A controls, but did increase the latency relative to F645A genotypes (*Figure 3C*). We conclude that ecdysone ligand-binding activity of EcR-A is required for nociception in third instar larvae.

TrpA1 knockout larvae lacking expression of all TrpA1 isoforms lose nociceptive behavior in response to thermal stimuli at 40–44°C, but are still able to respond to a 46°C probe with nociceptive behavior (*Gu et al., 2019*). Therefore, we sought to determine whether disruption of EcR function has effects specific to the probe temperature. Interestingly, EcR-A-W650A expression affected nociceptive behavior induced with the 42°C probe (*Figure 3B, C*) but not with the 46°C probe (*Figure 3D, E*). It thus appears that EcR-A coactivator assembly is not required for third instar nociception, while ligand-binding activity is required for third instar nociceptive behavior in response to a 42°C probe.

Having found that ecdysone ligand activity through EcR-A is required for nociception, we sought to determine if overexpression of EcR-A in C4da neurons is sufficient to render second instar larvae precociously nociceptive. Since second instar larvae have a low thermal nociceptive response rate, we assayed for a gain of thermal nociceptive response in second instar larvae with EcR-A overexpression in their C4da neurons. Indeed, overexpression of EcR-A was sufficient to increase thermal nociceptive behavior in second instar larvae (*Figure 4A, C*). In contrast to the temperature specific effect of dominant negative mutant expression in the third instar, overexpression of EcR-A in the second instar was able to promote nociception at both 42 and 46°C probe temperatures (*Figure 4A, C, E*). Of the larvae which did display a nociceptive behavior, overexpression of EcR-A did not significantly change the latency of the nociceptive response as compared to second instar control larvae (*Figure 4B*).

Previous studies have found that overexpression of each EcR isoform can suppress expression of the other EcR isoforms (*Schubiger et al., 2003*). Additionally, overexpression of each EcR isoform could rescue loss of EcR function mutations in specific tissues (*Cherbas et al., 2003*). The difference between EcR isoforms is within the A/B domains (*Figure 2—figure supplement 1A*). Therefore, we tested whether nociception in second instar larvae could be increased by overexpression in C4da neurons of EcR-A, EcR-B1, or EcR-ΔC. EcR-ΔC is an artificial isoform which contains only the sequences common to all isoforms and no A/B domain. We found that neither overexpression of EcR-B1 nor overexpression of EcR-ΔC could increase nociception of second instar larvae (*Figure 4A*). Additionally, overexpression of these EcR isoforms did not alter nociception in the third instar (*Figure 4—figure supplement 2A, B*). Given that expression of neither the domains common to all EcR isoforms, which are present in EcR-ΔC, nor the AF1 domain of EcR-B1 could increase nociception in the second instar, we conclude that the A/B domain unique to EcR-A is required to increase nociception of second instar larvae.

Expression of EcR-A RNAi or EcR-B1-dominant negative mutant in the third instar larvae reduces the number of dendrite tips (*Ou et al., 2008*; *McParland et al., 2015*). Insensitivity to nociception is often associated with a reduction in dendrite structure, while hypersensitivity to nociception can be

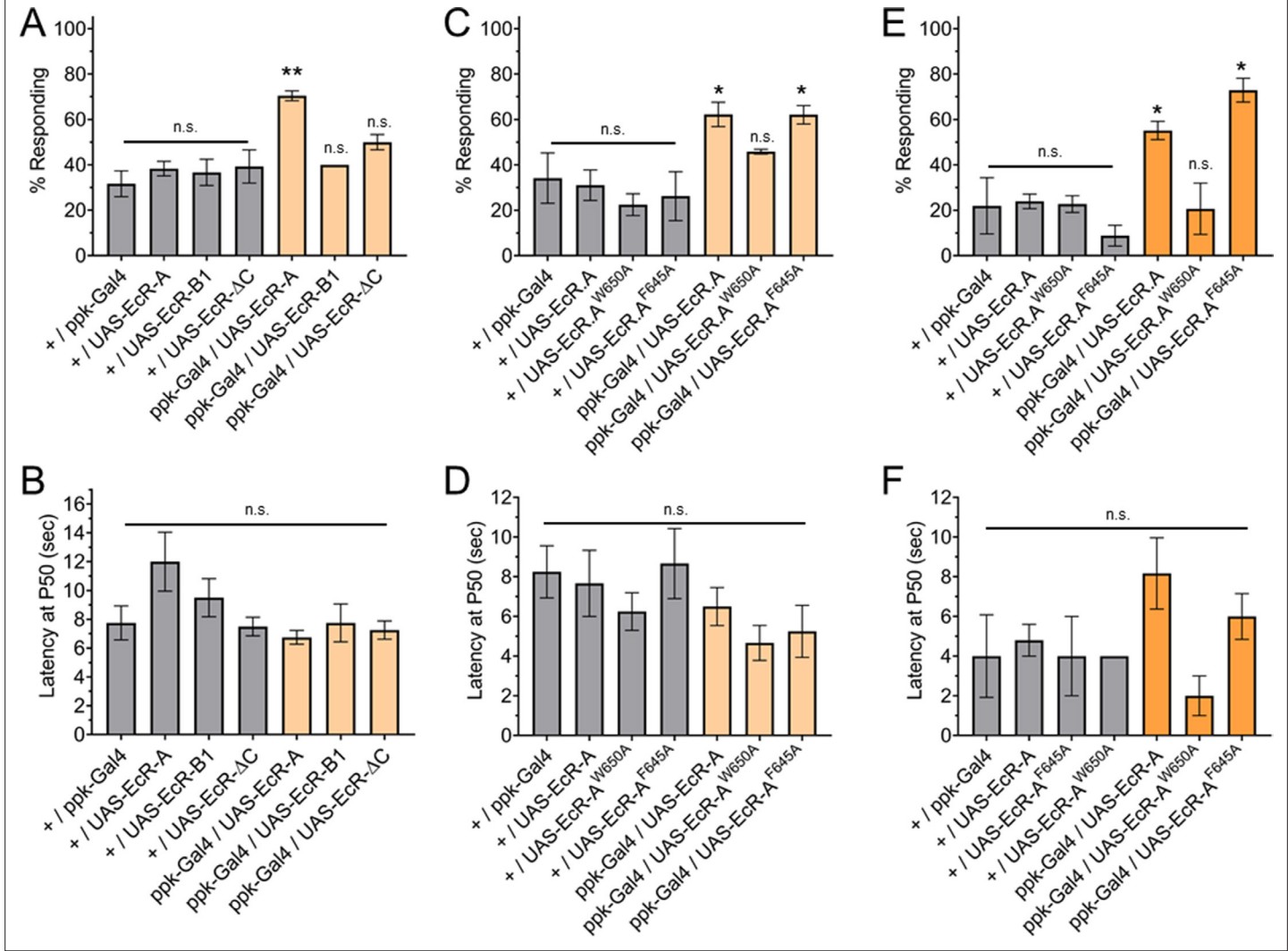

**Figure 4.** Ecdysone receptor (EcR)-A overexpression in Class IV dendritic arborization (C4da) neurons increases thermal nociception in the second instar. (**A**) Nociceptive behavior in second instar larvae with EcR isoforms overexpressed in C4da neurons. (**C, E**) Nociceptive behavior in second instar larvae with overexpression of EcR-A mutants. Ligand-binding mutant EcR-A (ppk-Gal4/UAS-EcR-W650A), or coactivator mutant EcR-A (ppk-Gal4/UAS-EcR-F645A). (**B, D, F**) Second instar latency by which 50% of the responding population displaying nociceptive behavior when overexpressing wild-type or mutant EcR-A. (**A–D**) 42°C nociceptive probe and (**E, F**) 46°C nociceptive probe. One-way analysis of variance (ANOVA) with Tukey's post test. *p < 0.05, **p < 0.01. *n* = 2–4 staging replicates of 15–20 larvae were tested for each genotype.

The online version of this article includes the following figure supplement(s) for figure 4:

**Figure supplement 1.** Ecdysone receptor (EcR)-A overexpression in Class IV dendritic arborization (C4da) neurons reduces dendrite tip number in the second instar.

**Figure supplement 2.** Ecdysone receptor (EcR) isoform overexpression in Class IV dendritic arborization (C4da) neurons does not change thermal nociception in the third instar.

associated with both increased and decreased dendrite coverage (*Honjo et al., 2016*). We examined whether the EcR-A overexpression induced nociception was associated with changes of the dendrite branch number or arbor size. We found that EcR-A overexpression in the second instar reduced the number of dendrite tips (a measure of branch number) without significantly changing the area of the dendrite arbor (*Figure 4—figure supplement 1A–D*). Our observation of decreased dendrite tip number with an associated increase in nociception is consistent with previously observed trends of decreased dendrite coverage in other nociceptive hypersensitive phenotypes (*Honjo et al., 2016*), further adding to the evidence that dendrite structure alone cannot predict nociceptive sensitivity.

Having found that ecdysone binding to EcR-A is required for third instar nociception, we sought to determine whether ligand binding is required for the precociously induced second instar nociception. We found that with overexpression of EcR-A-F645A, larvae were still precociously nociceptive as second instar larvae and they responded at a similar level as those overexpressing wild-type EcR-A (*Figure 4C*), suggesting that coactivator recruitment is not required for EcR-A induction of thermal nociception. In contrast, overexpression of the ligand-binding mutant EcR-A-W650A did not facilitate precocious development of nociceptive behavior in second instar larvae to the level exhibited by larvae overexpressing wildtype EcR-A (*Figure 4C*). These effects of EcR mutant expression were observed at both 42 and 46°C probe temperatures (*Figure 4C, E*). Of those larvae which did display a nociceptive behavior, overexpression of mutant EcR-A did not significantly change the latency of the nociceptive response (*Figure 4D, F*). These findings reveal that thermal nociceptive development in the second instar larvae is sensitive to ecdysone regulation mediated by EcR-A via its ligand-binding domain.

Together, these results demonstrate that EcR-A signaling is necessary and sufficient for the development of thermal nociceptor behavior. EcR-A acts in a cell autonomous manner in the C4da sensory neurons and requires the function of the ligand-binding domain. Our results also suggest that coactivator recruitment is not required for thermal nociceptive development or maintenance. Additionally, our results suggest that in the third instar, the maintenance of ability to detect 46°C is independent of EcR-A ligand binding.

## EcR regulates *subdued* expression during larval thermal nociceptive development

Having found that the EcR-A A/B domain and ligand binding are required for development of thermal nociception, we sought to determine whether any of the following nociceptive sensors are transcriptionally regulated by EcR-A. TrpA1 and Painless are transient receptor potential (TRP) channels required for thermal nociceptive behavior and nociceptive temperature-induced $Ca^{2+}$ activity in C4da neurons (*Tracey et al., 2003*; *Sokabe et al., 2008*; *Zhong et al., 2012*; *Terada et al., 2016*; *Gu et al., 2019*). The TMEM16 family member Subdued is also required for thermal nociceptive behavior (*Jang et al., 2015*). To assess the scope of EcR-A regulation, we also included the degenerin/epithelial sodium channel (DEG/ENaC) channel Ppk1 that is required for ambient temperature-driven behavior (*Ainsley et al., 2008*) and also functions along with Ppk26 in mechanical nociception (*Gorczyca et al., 2014*; *Mauthner et al., 2014*), as well as Piezo, which is involved in mechanical nociception but not thermal sensation in C4da neurons (*Kim et al., 2012*).

In order to compare the changes in gene expression during the sensory switch in larval development, we conducted quantitative real-time PCR (qRT-PCR) with the nociceptive genes in GFP-labeled C4da neurons isolated by fluorescence-activated cell sorting (FACS) from control second and third instar larvae. We found decreased expression of *TrpA1*, *subdued*, *ppk1*, and *ppk26* in third instar when compared to second instar C4da neurons. In contrast, *painless* expression increased while *piezo* expression did not significantly change from second to third instar C4da neurons (*Figure 5A*).

Next, we overexpressed EcR-A in C4da neurons and isolated these neurons with FACS from second instar larvae to measure expression of the nociceptive sensors. Comparing transcript levels of EcR-A-overexpressing C4da neurons to controls revealed that *subdued* expression was suppressed in EcR-A expressing C4da neurons as compared to that in control second instar C4da neurons (*Figure 5B*), thus mimicking the decrease during developmental transition from second to third instar. Notably, *subdued* stood out among the channel genes tested as the only one that displayed EcR-A regulation consistent with the developmental sensory switch.

While overexpression of EcR-A increased nociception in second instar larvae, overexpression of the ligand-binding mutant EcR-A-W650A did not increase nociception in second instar but reduced nociception in the third instar (*Figures 3 and 4*). Having found that EcR-A overexpression specifically suppressed *subdued* expression, we sought to determine whether expression of the EcR-A-dominant negative mutant has any effect on the level of *subdued* expression. When we overexpressed EcR-A-W650A in C4da neurons and then used FACS to isolate these neurons from either second or third instar larvae, we observed a reduction of *subdued* expression at both stages of larval development (*Figure 5—figure supplement 1A, B*). Additionally, we found that expression EcR-A-W650A in second instar larvae significantly reduced *TrpA1*, *ppk26*, and *piezo* expression (*Figure 5—figure supplement*

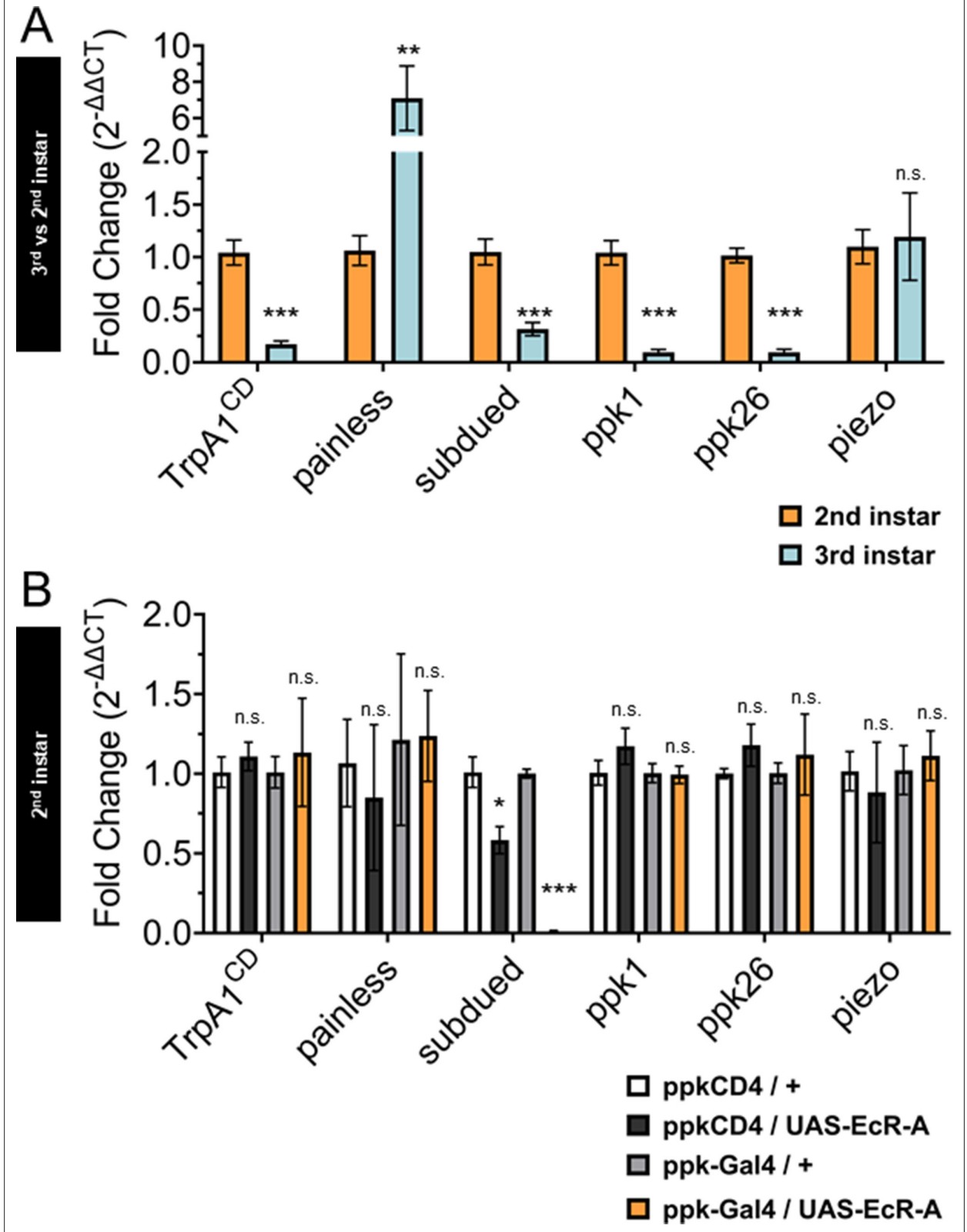

**Figure 5.** Ecdysone receptor (EcR) transcriptionally regulates *subdued* during the period of thermal nociceptive development. Expression of nociceptive genes measured by qRT-PCR from fluorescence-activated cell sorting (FACS) purified Class IV dendritic arborization (C4da) neurons. (**A**) C4da neurons isolated from second and third instar larvae. Expression in third instar neurons was normalized to second instar expression. (**B**) C4da neurons isolated from second instar larvae expressing wild-type EcR-A. Genotypic controls not expressing EcR-A (ppkCD4/UAS-EcR-A) were normalized

*Figure 5 continued on next page*

*Figure 5 continued*

to age-matched neurons (ppkCD4/+). Expression in EcR-A expressing neurons (ppk-Gal4/UAS-EcR-A) was normalized to age-matched control neurons without EcR-A expression (ppk-Gal4/+). Student's *t*-test. *p < 0.05, **p < 0.01, ***p < 0.001. (A) *n* = 8 and (B) *n* = 3–4 FACS isolation/staging replicates for each age and genotype.

The online version of this article includes the following figure supplement(s) for figure 5:

**Figure supplement 1.** Mutant ecdysone receptor (EcR) transcriptionally regulates nociceptive genes during the period of thermal nociceptive development.

**Figure supplement 2.** Ecdysone receptor A (EcR-A) RNAi reduces EcR-A protein but does not alter nociceptive gene expression.

*1A*), while EcR-A-W650A expression in the third instar larvae significantly reduced *ppk1*, *ppk26*, and *piezo* expression (*Figure 5—figure supplement 1B*).

Because *EcR-A* RNAi expression in C4da neurons has been shown to reduce nociception in the third instar, we also measured expression of the nociceptive sensors from FACS isolated third instar C4da neurons which expressed *EcR-A* RNAi. Expression of the *EcR-A* RNAi robustly reduced the presence of EcR-A protein in C4da neurons (*Figure 5—figure supplement 2A*), but we found no significant difference in the expression of nociceptive sensor genes (*Figure 5—figure supplement 2B*).

These data demonstrate that EcR-A regulates *subdued* expression. EcR-A overexpression can specifically suppress *subdued* expression in second instar larvae, as expected from the decrease of *subdued* expression in third instar larvae. We also found that inhibiting EcR-A ligand-binding capacity with expression of EcR-A-W650A can reduce expression of more nociceptive genes than expression of wild-type EcR-A. EcR-A-W650A expression reduced *subdued* expression in the second instar, and further reduced expression of *subdued* in the third instar larvae. We conclude that during the developmental period of the sensory switch, EcR-A regulates expression of *subdued*, and that ligand-binding activity is necessary for maintaining the regulated expression of multiple nociceptor genes in the third instar.

## Reduction of *subdued* confers second instar nociceptors with thermal and light sensitivity characteristic of third instar nociceptors

Our transcriptional data indicate that EcR-A overexpression specifically suppresses *subdued* expression in second instar C4da neurons. Therefore, we sought to determine whether *subdued* expression in second instar C4da neurons is responsible for the suppression of thermal nociceptive behavior in early larval development.

A knockout mutant of the *subdued* gene locus (*subdued^KO11*) (*Wong et al., 2013*) eliminated *subdued* mRNA expression (*Figure 6—figure supplement 1A*), but did not exhibit a significant change of nociceptive behavior in second instar larvae (*Figure 6A*). However, *subdued^KO11* larvae were significantly smaller than age-matched second instar control larvae (*Figure 6—figure supplement 1B*), raising the possibility that pleiotropic effects of *subdued* loss of function mutation in the whole animal may alter the behavioral dynamics and mask the effect of mutation in the C4da neurons. Expression of a *subdued*-RNAi was effective at reducing *subdued* expression but did not eliminate all expression (*Figure 6—figure supplement 1C*). Therefore, we tested the effect of expression of *subdued*-RNAi in C4da neurons, which did not reduce the size of second instar larvae (*Figure 6—figure supplement 1D*). When *subdued*-RNAi was expressed specifically in the C4da neurons, second instar larvae had a greater nociceptive behavioral response than controls (*Figure 6B*). Thus, RNAi reduction of *subdued* expression in the C4da neurons caused a precocious thermal nociceptive response in second instar larvae, phenocopying EcR-A overexpression.

Having found that UV light-induced $Ca^{2+}$ activity in C4da neurons decreased during larval development, we sought to determine whether Subdued is involved in the reduction of UV-A $Ca^{2+}$ activity from second to third instar. We observed that expression of *subdued*-RNAi decreased both the number of actively responding neurons and the peak $Ca^{2+}$ activity in C4da neurons during light stimulation of second instar larvae (*Figure 6C–E*).

Together these data demonstrate that reducing expression of *subdued* in second instar larvae is sufficient not only to increase thermal nociception but also to decrease the UV-A light response, thus enabling second instar C4da neurons to respond to thermal as well as light stimuli in the same manner as third instar C4da neurons. These effects of reduced *subdued* expression match the developmental

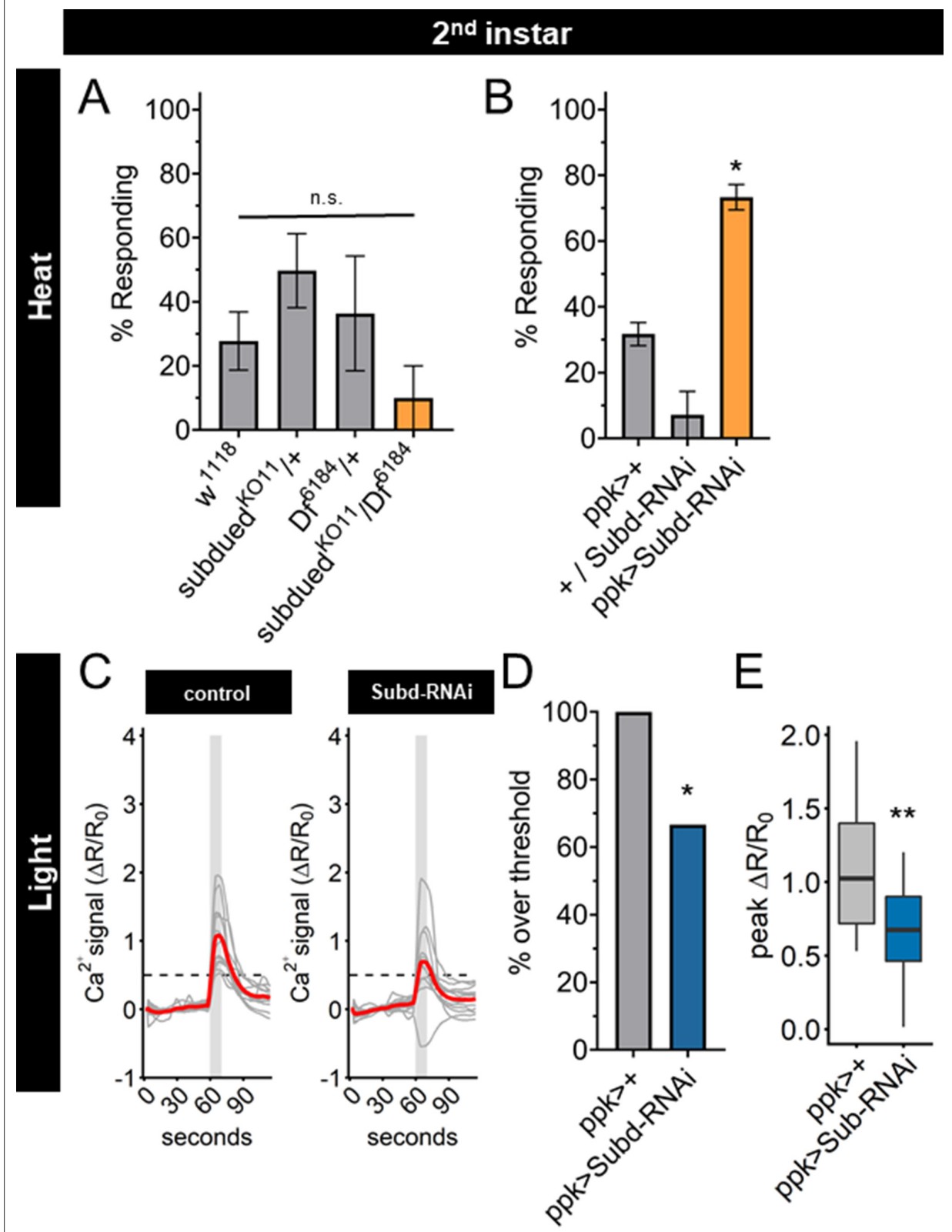

**Figure 6.** Subdued-RNAi changes second instar Class IV dendritic arborization (C4da) neuron sensory response pattern to third instar sensory response pattern. (**A**) Second instar larvae with 42°C nociceptive probe. Percent of population displaying nociceptive behavior when mutant for Subdued. (**B**) Second instar larvae with 42°C nociceptive probe. Percent of population displaying nociceptive behavior when expressing Subdued-RNAi (ppk-Gal4 > Subd RNAi v37472). (**C**) Individual traces (gray lines) and means (red lines) of Ca²⁺ activity calculated by ratiometric change from baseline. Gray column

*Figure 6 continued on next page*

*Figure 6 continued*

indicates period of stimulus. Dotted line indicates threshold level for % over threshold calculations. (**D**) Percent of soma with Ca$^{2+}$ activity over the threshold level. (**E**) Peak Ca$^{2+}$ activity during the periods of stimulus programs. (A, B) One-way analysis of variance (ANOVA). (**D**) Fisher's exact test, one-sided. (**E**) Student's *t*-test. (A, B) $n = 3–4$ staging replicates of 15–20 larvae. (C–E) ppk > $+n = 14$ neurons, ppk > Subd RNAi $n = 12$ neurons. *$p < 0.05$, **$p < 0.01$.

The online version of this article includes the following figure supplement(s) for figure 6:

**Figure supplement 1.** Subdued mutants have reduced larvae growth during the second instar.

progression of the sensory switch. We conclude that EcR-A regulation of the C4da sensory switch is mediated in part by reduction of *subdued* expression levels: *subdued* expression, owing to the EcR-A unliganded activity in second instar C4da neurons, is associated with a lack of thermal nociception, whereas suppression of *subdued* expression via EcR-A overexpression or *subdued*-RNAi in second instar C4da neurons causes precocious thermal nociception while reducing responsiveness to UV light (***Figure 6—figure supplement 1E***).

## Discussion

The impact of peripheral nervous system (PNS) development on behavior is underscored by the recent finding that defects in the PNS are sufficient to produce symptoms in mouse models of autism spectrum disorders (***Orefice et al., 2016***). The role of hormonal regulation of sensory systems during periods of postnatal development has also been highlighted by studies of growth hormone regulation of thermal sensitivity (***Liu et al., 2017***; ***Ford et al., 2019***). Considering that sensory neurons progress through transcriptionally distinct identities during development (***Sharma et al., 2020***), characterizing how the specific changes in receptors or channels that produce these developmentally specific outcomes is of prominent importance for understanding the development of neurological disease.

In this study, we address how a nociceptive behavior is temporally acquired during development. We demonstrate how steroid hormone regulation of nociceptor activity in a class of sensory neurons in the PNS adjusts the developing larval response to nociceptive stimuli. Heat-induced nociceptive behavior in *D. melanogaster* larvae arises during the final (third) instar of development (***Sulkowski et al., 2011***). Here, we show that this behavioral transition reflects a nociceptive sensory switch through regulation of Subdued by the steroid hormone ecdysone and the EcR isoform EcR-A.

While previous studies have focused on EcR localization in the third instar and pupal stages of C4da neurons, we show EcR is present in both the nucleus and cytoplasm in the second instar, followed with an increase of EcR nuclear localization 8 hr into the third instar (***Figure 2***). Previous work has shown that loss of EcR-A reduces thermal nociception in the third instar (***McParland et al., 2015***). While we did not find detectable transcriptional changes in nociceptor genes in third instar C4da neurons during *EcR-A* RNAi expression, we found broad suppression of nociceptor genes during EcR ligand-binding mutant (W650A) expression (***Figure 5—figure supplement 2***). Through behavioral and transcriptional analysis, we show that EcR-A nociception regulation is ligand dependent, by demonstrating that increased ecdysone titer and overexpression of ligand-competent EcR both accelerate nociception in the second instar (***Figures 3 and 4***). Ligand binding of EcR leads to widespread epigenetic changes through release of corepressors and recruitment of coactivators (***Uyehara et al., 2017***; ***Uyehara and McKay, 2019***). The precocious activation of nociception in the second instar requires the unique A/B domain of the isoform EcR-A, as other isoforms lacking this domain do not increase nociception (***Figure 4A***). The A/B domain of EcR-A is less activating and more repressive then the EcR-B1 A/B domain (***Dela Cruz et al., 2000***; ***Mouillet et al., 2001***; ***Hu et al., 2003***). It thus appears that ligand-dependent derepression could be the action which promotes nociception. Moreover, expression of the EcR coactivator mutant (F645A) phenocopies expression of wild-type EcR in its ability to promote nociception (***Figure 4B***), suggesting that in contrast to coactivator recruitment, preservation of the ability to remove corepressors is required for nociception. This regulation of repression may account for the ability of the EcR ligand-binding mutant (W650A) to suppress multiple nociceptive genes including *subdued* in the second and third instars. The release of corepressors could allow other transcription factors to gain access to response elements which suppress *subdued*, or directly promote transcription of a repressor of *subdued*. How derepression leads to the suppression of *subdued* is an intriguing open question.

For third instar larval nociception, our data suggest there are both ligand-dependent and -independent EcR-A pathways. We find third instar expression of EcR-A ligand-binding mutant to have a temperature-specific phenotype. Expression of the EcR-A ligand-binding mutant inhibits nociception at 42°C, but does not reduce nociception at 46°C (*Figure 3B, D*). This temperature-specific effect is reminiscent of the phenotype of TrpA1 mutant larvae, which lose nociceptive behavior at probe temperatures of 44°C and lower, but are still responsive to 46°C stimuli (*Gu et al., 2019*). We found that expression of EcR-A ligand-binding mutant reduced *subdued* expression in the third instar even more than during the developmental transition (*Figure 5A, C*), suggesting that this loss of *subdued* expression may contribute to the phenotype at 42°C but not at 46°C. Consistent with this possibility, we observed that Subdued-RNAi was able to increase nociception in second instar larvae while Subdued knockout mutation did not increase nociception. Together these results suggest that the amount of *subdued* expression is important for promoting or inhibiting nociception. Furthermore, there are Subdued and EcR-A ligand-independent pathways which promote nociception in the third instar. We observe alteration of *TrpA1*, *painless*, *ppk1* and *ppk26* expression during development but not as a result of EcR overexpression (*Figure 5*). These separate pathways may involve mechanisms of regulating temperature specificity in C4da neurons.

Subdued has homology to the Calcium-activated Chloride Channels (CaCC) of the mammalian TMEM16 family, with channel activity similar to both TMEM16A and TMEM16F (*Wong et al., 2013*; *Le et al., 2019*). The effect of CaCC on neural activity is dependent on the intracellular Cl$^-$ concentration ([Cl$^-$]$_i$) (*Berg et al., 2012*). When [Cl$^-$]$_i$ is high, elevation of internal Ca$^{2+}$ level activates CaCCs and causes Cl$^-$ efflux, leading to excitation. In contrast, when [Cl$^-$]$_i$ is low, Ca$^{2+}$ activation of CaCCs causes an influx of Cl$^-$ which has an inhibitory effect on neuronal excitability. Differential expression of symporters and cotransporters in the CNS and PNS, as well as in immature and mature neurons, is important for creating the inhibitory or excitatory effects of CaCCs. In third instar larvae, Cl$^-$ efflux has been observed during thermal nociceptive stimulation of C4da neurons (*Onodera et al., 2017*). However, Onodera et al. found that loss of Subdued did not significantly change the magnitude of the Cl$^-$ efflux during heat stimulus, suggesting other channels are responsible for the Cl$^-$ efflux in third instar C4da neurons. The direction of Cl$^-$ movement in the second instar remains to be determined, and it is conceivable that Subdued could produce an inhibitory influx of Cl$^-$ in the second instar. Characterizing Cl$^-$ homeostasis during C4da neuron development and its impact on nociception will be of interest in future studies.

Subdued has previously been found to regulate thermal nociception in third instar larvae (*Jang et al., 2015*). Jang et al. found the strongest decrease of nociception in the third instar when expression of *subdued* was reduced in all dendritic arborization neurons. Additionally, when *subdued* was overexpressed in C4da neurons the number of third instar larvae that responded to nociceptive heat did not significantly change. Interesting, when *subdued* was overexpressed with a subdued-Gal4, the number of fast responding third instar larvae increased (*Jang et al., 2015*). This suggests there may be a substantial thermal nociception role for Subdued in dendritic arborization neurons other than C4da neurons during third instar. Our data suggest that as C4da neurons develop from second to third instars, the ability of Subdued to influence detection of stimuli may change. We find that thermal nociception is enhanced by reduction of *subdued* expression in second instar C4da neurons, via either EcR-A overexpression or Subdued-RNAi knockdown. Additionally, we find that expression of the EcR-A-ligand mutant reduces *subdued* expression even further in third instar C4da neurons, and that this is associated with a loss of nociception. The greater amount of *subdued* expression in the second instar may mean that Subdued could have a greater impact on Cl$^-$ regulation during the second instar. This mechanism of Subdued activity would be dependent on the intracellular Cl$^-$ concentration created by symporters and cotransporters. The importance of Cl$^-$ regulation has recently been highlighted in the Class III da neurons, as both Subdued and depolarizing Cl$^-$ currents have recently been found to promote cold-evoked nociception (*Himmel et al., 2021*). Therefore, identifying regulators of Cl$^-$ concentration during the development of C4da neurons and their impact on nociception will be of interest in future studies.

Our work adds to the evidence that the developmental transition of thermal nociception represents a shift in sensory state rather than a change in sensory system construction: (1) optogenetics can stimulate aversive behavior in both second and third instar C4da neurons (*Sulkowski et al., 2011*), (2) TrpA1 misexpression can confer nociceptive behavior in both second and third instar larvae exposed

to innocuous temperatures (*Luo et al., 2017*), (3) UV-A stimulation induces greater Ca²⁺ response in second instar than in third instar C4da neurons (*Figure 1*), and (4) reduction of *subdued* expression in second instar renders C4da neurons responsive to thermal nociceptive stimuli (*Figure 6*). Additional properties of C4da neurons during this developmental transition remain to be determined. These properties include: whether thermally induced calcium activity in C4da neurons (*Terada et al., 2016*; *Gu et al., 2019*) also changes over this period of development, whether EcR regulates the 'burst and pause' encoding (*Terada et al., 2016*), and whether EcR regulation alters synaptic connections of C4da neurons (*Valdes-Aleman et al., 2021*).

Recent findings have highlighted the occurrence of developmentally timed modulation of sensory state in both larvae and adult *Drosophila*. A sensory switch in thermotaxis behavior of *Drosophila* larvae is regulated by transcriptional regulation of thermoreceptors (*Tyrrell et al., 2021*). In adults, male courtship behavior is modulated through olfactory neuron pheromone sensitivity and hormone-mediate chromatin reprograming (*Zhao et al., 2020*). Our study highlights that temporally programed sensory switches may be a widely used developmental mechanism to match behavioral outcomes with life history events.

We speculate that the nociceptive shift in sensory state is an important mechanism for matching sensory system function with life history transitions. *D. melanogaster* larvae do not develop robust thermal nociception until the second half of the larval phase. This means that for the first half of larval life, larvae do not have thermal nociception behavior. In contrast, nociceptor activity to UV-A light decreases from the second to third instars, suggesting that nociceptive sensitivity to short wavelength light decreases during the second half of the larval phase. This transition correlates with larval behavioral changes that emerge during the third instar, such as cessation of feeding beneath the surface and wandering above ground (*Wegman et al., 2010*). Larvae are transitioning from predominantly feeding behavior in the second instar, often shielded from light and high temperatures, to movement away from food in search of pupation sites in the third instar. In a natural habitat, this transition can mean leaving the food source, which necessitates greater exposure to heat, increasing the risk of predation and desiccation (*Ballman et al., 2017*). This sensory switch may function as a developmental strategy, shaping behavioral outcomes as animals encounter changing environments, thus increasing survival.

## Materials and methods

### *D. melanogaster* stocks and culturing

Detailed stock genotypes and sources are listed in *Table 1*. X chromosome genotypes are simplified: male and female larvae are pooled together in test populations, and the origin of miniwhite alleles could be mixed. Control genotypes are crossed to w[1118] (v60000) unless otherwise noted.

All experimental crosses were raised at 25°C and 60% humidity, with a 12 hr light–dark cycle, and fed cornmeal-molasses media. Larval development was synchronized by staged egg collection on grape agar, followed by synchronized collection of newly hatched L1 larvae. Second instar larvae were assayed between 26 and 32 hr after L1 larval hatching and third instar larvae were collected as prepupariation wandering larvae. All assays were performed during the dark–light cycle.

### Nociceptive behavior assay

Nociceptive stimulation was performed as previously described (*Babcock et al., 2009*). Larvae were briefly rinse in dH₂0 and allowed to acclimatize for 1 min on a vinyl sheet. The larvae were kept moist and a temperature-controlled heat probe (ProDev Engineering, TX) was used to apply heat between the larval body segments A3–A6 of forward crawling larvae. Nociceptive behavior was defined as at least one complete 360° roll. The time to complete one roll (latency) was measured up to a 20 s cutoff. Each larva was presented with a single stimulus to avoid habituation. Latency was calculated for each biological replicate as the time at which greater than or equal to 50% of the responding population (nonresponding larvae we excluded) had responded to the stimulus.

### 20E feeding

For food supplemented with 20E (Sigma, H5142), 20E was dissolved in 100% ethanol and mixed with room temperature cornmeal-molasses media. An equivalent volume of ethanol alone was mixed with

**Table 1.** *Drosophila melanogaster* stocks used in this study.

| Figure | Genotype | Source |
|---|---|---|
| 1A and S1-1 | ppk-tdGFP[1b],UAS-Dcr2/+;ppk-Gal4[1a]/+ | *Han et al., 2011* |
| 1C–F | ppk-Gal4[vk37]/+;UAS-GCaMP6(s),UAS-tdTom/+ | BDSC 42749, *Han et al., 2011* |
| 2A–C, S2-1B, 3A | ppk-tdGFP[1b],UAS-Dcr2/+;ppk-Gal4[1a]/+ | |
| 3B and S3-1A–C | ppk-tdGFP[1b],UAS-Dcr2/+;ppk-Gal4[1a]/+ | |
| | P{UAS-EcR.A.W650A}TP5/+ | BDSC 9451 |
| | P{UAS-EcR.A.F645A}TP2/+ | BDSC 9452 |
| | ppk-tdGFP[1b],UAS-Dcr2/P{UAS-EcR.A.F645A}TP2;ppk-Gal4[1a]/+ | |
| | ppk-tdGFP[1b],UAS-Dcr2/P{UAS-EcR.A.W650A}TP5;ppk-Gal4[1a]/+ | |
| 4A and S4-2A, B | ppk-Gal4[vk37] | *Han et al., 2011* |
| | P{UAS-EcR.A}3a/+ | BDSC 6470 |
| | P{UAS-EcR.B1}3b/+ | BDSC 6469 |
| | P{UAS-EcR.C}TP1-4/+ | BDSC 6868 |
| | ppk-Gal4[vk37];P{UAS-EcR.A}3a | |
| | ppk-Gal4[vk37]; P{UAS-EcR.B1}3b | |
| | ppk-Gal4[vk37];P{UAS-EcR.C}TP1-4 | |
| 4B and S4-1A–G | ppk-tdGFP[1b],UAS-Dcr2/+;ppk-Gal4[1a]/+ | |
| | P{UAS-EcR.A}3a/+ | |
| | P{UAS-EcR.A.F645A}TP2/+ | |
| | P{UAS-EcR.A.W650A}TP5/+ | |
| | ppk-tdGFP[1b],UAS-Dcr2/+;ppk-Gal4[1a]/P{UAS-EcR.A}3a | |
| | ppk-tdGFP[1b],UAS-Dcr2/P{UAS-EcR.A.F645A}TP2;ppk-Gal4[1a]/+ | |
| | ppk-tdGFP[1b],UAS-Dcr2/P{UAS-EcR.A.W650A}TP5;ppk-Gal4[1a]/+ | |
| 5A | ppk-tdGFP[1b],UAS-Dcr2/+;ppk-Gal4[1a]/+ | |
| 5B | ppk-CD4- tdGFP[1b] /+ | |
| | ppk-CD4- tdGFP[1b] /+;P{UAS-EcR.A}3a/+ | |
| | ppk-tdGFP[1b],UAS-Dcr2/+;ppk-Gal4[1a]/+ | |
| | ppk-tdGFP[1b],UAS-Dcr2/+;ppk-Gal4[1a]/P{UAS-EcR.A}3a | |
| S5-1A, B | ppk-CD4- tdGFP[1b] /+ | |
| | ppk-CD4- tdGFP[1b] / P{UAS-EcR.A.W650A}TP5 | |
| | ppk-tdGFP[1b],UAS-Dcr2/+;ppk-Gal4[1a]/+ | |
| | ppk-tdGFP[1b],UAS-Dcr2/P{UAS-EcR.A.W650A}TP5;ppk-Gal4[1a]/+ | |
| S5-2A, B | ppk-tdGFP[1b],UAS-Dcr2/+;ppk-Gal4[1a]/+ | |
| | ppk-CD4- tdGFP[1b] /+;P{w[ + mC] = UAS EcR.A.dsRNA}91/+ | BDSC 9328 |
| | ppk-tdGFP[1b],UAS-Dcr2/+;ppk-Gal4[1a]/ P{w[ + mC] = UAS EcR.A.dsRNA}91 | |

*Table 1 continued on next page*

*Table 1 continued*

| Figure | Genotype | Source |
|---|---|---|
| 6A, S6-1A, B | Subdued[KO11]/+ | *Wong et al., 2013* |
| | Df(3R)Exel6184, P{w[ + mC] = XPU}Exel6184 | BDSC 7663 |
| | Subdued[KO11]/Df(3R)Exel6184, P{w[ + mC] = XPU}Exel6184 | |
| 6B, S6-1D | ppk-tdGFP[1b],UAS-Dcr2/+;ppk-Gal4[1a]/+ | |
| | P{GD3674}v37472/+ | VDRC 37472 |
| | ppk-tdGFP[1b],UAS-Dcr2/+;ppk-Gal4[1a]/ P{GD3674}v37472 | |
| S6-1C | P{GD3674}v37472/+ | VDRC 37472 |
| | Actin-Gal4/+; Subdued[KO2]/+ | *Wong et al., 2013* |
| | Actin-Gal4/+; Subdued[KO2]/P{GD3674}v37472 | |
| 6-1C–E | ppk-Gal4[vk37]/+;UAS-GCaMP6(s),UAS-tdTom/+ | BDSC 42749, *Han et al., 2011* |
| | ppk-Gal4[vk37]/+;UAS-GCaMP6(s),UAS-tdTom/ P{GD3674}v37472 | |

media as a control. Larvae were transferred to supplemented media 48 hr AEL and assayed for nociceptive behavior 8 hr later.

## Ultraviolet light response calcium imaging

C4da response to UV-A stimulation was measured as previously described (*Yadav et al., 2019*). A Leica SP5 confocal microscope with resonance scanner was used with the ×20 oil immersion objective and ×16 optical magnification. Imaging was done with 512 × 512 resolution and a slice thickness of 5 µm (20 Z slices) at 1.201 fps. A 405 laser line (50 mW) was used at 100% laser power. Neurons were imaged for 60 s before a 10-s UV exposure. Z-stacks were acquired in the GFP, RFP, an UV channels. The ratiometric signal was quantified as described for thermal $Ca^{2+}$ imaging.

## Cell purification and qRT-PCR

C4da neurons were isolated by dissecting 30–40 larvae in 1× phosphate-buffered saline (PBS) on ice. To increase C4da neuron concentration in the final cell suspension, the larval body wall was inverted and the CNS, imaginal discs, gut, and fatbody were removed. After dissection cell suspensions were prepared by vortexed with 1.5 µl 1× Liberase TM (Roche, LIBTM-RO) in 500 µl cold PBS. To dissociate cells samples were incubated for three periods at 25°C at 1000 rpm, triturating 10 times with a glass pipette between each incubation. Suspensions were strained through a 40-µm cell strainer (Fisher Scientific) and brought to 1.5 ml with Schneider's media. 1 µl ethidium homodimer-1 (Thermo Fisher, L3224) was added before sorting to mark dead cells. C4da neurons were isolated by FACS with an Aria II (Becton Dickinson). GFP+nonautofluorescent RFP− events were sorted into 20 µl lysis buffer (Thermo Fisher, KIT0204) and immediately frozen on dry ice. 100–1000 cells were isolated per dissection replicate. RNA was isolated with PicoPure RNA isolation kit (Thermo Fisher, KIT0204), cDNA was synthesized with Applied Biosystems High-Capacity cDNA Reverse Transcription Kit (Thermo Fisher, 4368813), and qRT-PCR was performed using SYBR green (Thermo Fisher, A25742) with a QuantStudio7 (Thermo Fisher). Relative expression was calculated using the detla-delta Ct method with the housekeeping gene eEF1α2 normalized to the mean Ct of the genotypic and isolation controls.

*Subdued* expression in knockout mutant and RNAi expressing larvae was measured using RNeasy Mini Kit (Qiagen 74104) isolation from three to four larvae per genotype. cDNA syntheses and qRT-PCR were performed as described FACS isolated cells.

## Primers

| Target | Forward | Reverse |
|---|---|---|
| TrpA1 isoform[CD] | GCCGGAACAGCAAGTATTGGA | CGATTTCAATCCGCTTGGGAC |
| painless | CAGTGAGCGACACCCAAGTTA | GAGAACCTGCTCGTACCGAC |
| subdued | GGAGGTCGAGTCCAGTCAGA | CTGGAATCTCCTTCATGGGCA |
| ppk1 | CCCGAAAAACGCCAATGTCT | ACAACCGCATTTGGAAACCG |
| ppk26 | GAGCGGAAGGTATTATTCCCGA | GGGAGATGTATCCGCACTGG |
| piezo | AAGCCACGGGTTTCTTTGC | GTTGGGGTGTACGTGCCTTT |
| eEF1α2 | CGTCTACAAGATCGGAGGCA | GACCATGCCTGGCTTGAGGA |

## Immunohistochemistry

Larval ages were staged by L1/L2 molt for second instar or L2/L3 molt for third instar. Larvae were filleted in PBS on ice and fixed for 10 min at room temperature in 4% paraformaldehyde. Samples were blocked with 5% goat serum for 2 hr at room temperature and then incubated with either EcR-common DDA2.7, EcR-A 15G1a, or EcR-B1 AD4.4 (Developmental Studies Hybridoma Bank) in blocking solution (1:200) overnight at 4°C. After rinse, samples were incubated with secondary antibody Alexa Fluor 555 (Invitrogen A-2142A) in blocking solution (1:500) for 2 hr at room temperature before 10 min staining with DAPI (1:10,000) and mounting in Vectashield (Vector Laboratories). Mean fluorescence was measured with Fiji (https://imagej.net/fiji) in traces of the nuclei and cytoplasm (traces of cytoplasm excluded C4da neuron nuclei and an neighboring nuclei) and adjusted for area before calculating the ratio of fluorescence in the nuclease and cytoplasm of C4da neurons.

## Larval size measurement

Larvae were collected in 75 µl PBS and placed in an 80°C thermo-block for 10 min. The turgid larvae were then placed on a microscope slide, imaged, and larval body area was measured with Fiji (https://imagej.net/fiji).

## Statistical analysis

Statistical tests were done with GraphPad Prism 8.3.0 (GraphPad Software) or R (version 3.5.2, R Core Team 2018).

## Acknowledgements

We thank Tun Li and Susan Younger for experimental consultation, Caitlin O'Brien, Han-Hsuan Liu, Ke Li, Maja Petkovic, Beverly Piggott, and Rebecca Jaszczak for editorial advice. Stocks obtained from the Bloomington *Drosophila* Stock Center (BDSC, NIH P40OD018537) and Vienna *Drosophila* Resource Center (VDRC, https://www.vdrc.at) were used in this study. DDA27 was obtained from the from the Developmental Studies Hybridoma Bank (DSHB). Research reported in this publication was supported by the National Institutes of Health: National Institute of General Medical Sciences F32GM130019 (JSJ) and National Institute of Neurological Disorders and Stroke R35NS097227 (YNJ). YNJ and LYJ are investigators at the Howard Hughes Medical Institute.

## Additional information

### Funding

| Funder | Grant reference number | Author |
|---|---|---|
| National Institute of General Medical Sciences | F32GM130019 | Jacob S Jaszczak |

| Funder | Grant reference number | Author |
|---|---|---|
| National Institute of Neurological Disorders and Stroke | R35NS097227 | Yuh Nung Jan |
| Howard Hughes Medical Institute | | Lily Yeh Jan Yuh Nung Jan |

The funders had no role in study design, data collection, and interpretation, or the decision to submit the work for publication.

## Author contributions

Jacob S Jaszczak, Conceptualization, Data curation, Formal analysis, Funding acquisition, Investigation, Methodology, Resources, Validation, Visualization, Writing - original draft, Writing - review and editing; Laura DeVault, Data curation, Investigation, Methodology, Validation, Writing - review and editing; Lily Yeh Jan, Funding acquisition, Writing - review and editing; Yuh Nung Jan, Funding acquisition, Methodology, Supervision, Writing - review and editing

## Author ORCIDs

Jacob S Jaszczak (iD) http://orcid.org/0000-0003-3752-1152
Laura DeVault (iD) http://orcid.org/0000-0001-6801-5098
Lily Yeh Jan (iD) http://orcid.org/0000-0003-3938-8498
Yuh Nung Jan (iD) http://orcid.org/0000-0003-1367-6299

## Decision letter and Author response

Decision letter https://doi.org/10.7554/eLife.76464.sa1
Author response https://doi.org/10.7554/eLife.76464.sa2

## Additional files

### Supplementary files
• Transparent reporting form

### Data availability

All data generated or analyzed during this study are included in the manuscript and supporting files.

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
