## [Editor Report]

The authors describe in vivo analyses of an intriguing steroid-mediated development shift in the sensitivity of *Drosophila* larvae to noxious stimulation as they move from the L2 to the L3 instar stage. Experiments and observations presented show that the steroid hormone ecdysone regulates nociceptor activity in the peripheral nervous system by suppressing expression of a gene named subdued, which encodes a membrane protein of the TMEM16 channel family.

---

## [Decision Letter]

**Decision letter after peer review:**

[Editors’ note: the authors submitted for reconsideration following the decision after peer review. What follows is the decision letter after the first round of review.]

Thank you for submitting your work entitled "Steroid hormone signaling activates thermal nociception during *Drosophila* peripheral nervous system development" for consideration by *eLife*. Your article has been reviewed by 3 peer reviewers, and the evaluation has been overseen by a Reviewing Editor and a Senior Editor. The reviewers have opted to remain anonymous.

Our decision has been reached after careful consultation between the reviewers. Based on these discussions and the individual reviews below, we genuinely regret to inform you that this work will not be considered further for publication in *eLife* in current form. However, if you choose to perform all of the large number of experiments requested by the reviewers, and if you so wish, then *eLife* should be happy to consider an extensively revised manuscript for review, nominally as a new submission.

The reviewers felt the problem being addressed was interesting and expressed particular interest in the role of subdued as a functional target of ECR, but had several concerns regarding both the clarity of underlying mechanisms as proposed in this manuscript and the overall level of advance over previous work (eg. McPharland et al.).

*Reviewer #1:*

This study by the Jan group examines the onset/development of functional thermal (and UV) nociception in *Drosophila* larvae. Previously, other groups had established that there are differences in nociceptive responses between second and third instar larvae (Cox) and that ecdysone receptor function is required for functional nociceptor development (Ganter). This study does a more thorough job than that previous work in documenting the stage differences and identifies a molecular player (subdued) that appears to be required for stage-specific changes in sensitivity. Overall, the data are sound and convincing and the paper is very well written. A few suggestions for improvement/clarification are offered.

1. Data organization. Some of the best data showing the stage-specific difference is in supplemental figure 1C- latency. This should really be in the main figure 1- probably as panel B. Figure 1C could benefit and would be more complete if second instar data were shown for 25 and 36 degrees.

2. The baseline calcium levels in second versus third instar is quite different- much higher in second instar looking at figure 1B. This data is essentially normalized/compared to same stage's baseline but this does raise the question: Does EcR function or subdued function shift the magnitude of response or shift the second instar (or third instar) baseline? It's a little hard to tell because Figure 3 (EcR) doesn't have stills of representative cells and Figure 6 (subdued, temperature) does not either. All we see here is the ratio data. Representative still data should be presented and whether the gene affects baseline calcium versus magnitude of change upon stimulus should be discussed.

3. Figure 2 organization: Personal preference but the 20E data currently in supplement 2B is probably more important to the main point than the third instar data (Main figure 2D).

4. Some of the discussion of the differences between second and third instar seems a bit oversimplified, especially as it refers to the EcR expression data in Figure 2 supplement. It's not really accurate to say that second instar has low EcR and third has high. Reality (from the data shown) is that nuclear EcR is high in early second and later third and low in late second and early third. There isn't really data to address intra-stage differences in responsiveness so this should be described a bit more clearly- without the current oversimplification. Adding modifiers (late second instar, early third instar) would help.

5. Analysis of subdued: Most helpful would be confirmation that this RNAi line is actually reducing subdued transcript levels, as one would predict. This would enhance confidence that there are no off-target effects. Also helpful would be the converse experiment to RNAi loss of function. Authors perform Subdued RNAi and see changes in second instar thermal responsiveness. Does overexpression of Subdued shift responsiveness in third instar?

*Reviewer #2:*

The manuscript deals with the basis of an intriguing shift in nociceptive behavior shown by *Drosophila* larvae as they move from the L2 to the L3 instar. Prior publications have shown that the shift is controlled by ecdysone and implicated the EcR-A knock-down of EcR-A levels rendered the neurons hyposensitive to noxious thermal and mechanical stimulation and also altered features of their anatomy. Reduction in EcR-B1 did not effect their responses but has anatomical ramifications. This paper extends these studies into Ca^2+^ data from specific da neurons and also reveals a gene, subdued, that may play a role in the change in responsiveness.

There are a number of substantial concerns:

1. The paper focuses on the differences between the L2 and L3 instars. According to the Methods, their L2 stage is in the middle of the L2 period, but the L3 stage is a wandering L3 larva which is at least 50 hours older. There are three critical ecdysone-related events that occur in this 50 hour span: the critical weight at about 8-10 hr after L3 ecdysis, the mid-L3 transition, and the ecdysone peak that initiates wandering. If the transition occurs during the transition from L2 to L3, then the "critical weight" transition would be a good candidate and L3 larvae about 12 to 18 hr after ecdysis would be more appropriate.

2. Another troubling methodological issue is that their behavioral tests were run at either 42 or 46 C while the Ca^2+^ measurements are done at 44 C. This becomes a concern in a situation like in Figure 2 where W650A expression is behaviorally effective at 42 C but has no effect at 46 C. We are then provided with a Ca^2+^ response at 44 C. Since this temp is half-way between the two used for the behavior tests, we have no way of assessing whether or not the Ca^2+^ response tracks the behavioral response.

3. A serious interpretational issue has to do with ignoring the complexity imposed by the EcR isoforms. Through most of the paper and figures, the authors speak of "EcR" when they really should be saying "EcR-A". This is not a trivial issue because the EcR-B isoforms have a strong hormone-independent transactivation domain at their N-terminus whereas EcR-A does not. At times when EcR-B1 is a present, the expression of wildtype EcR-A dampens the cellular response by competing with the endogenous EcR-B1 [Schubiger et al., 2003, Mech. Dev. 120:909]. Previous experiments (McPharland et al.) used RNAi to selectively knockdown different isoforms and these suggested a function for EcR-A. Ectopic expression of EcR-A alone, without EcR-B1 for comparison, can lead to wrong interpretations, especially since both isoforms are found in the da neurons (see McPharland).

4. Another problem is with the interpretation of the EcR dominant negatives (EcR-DN). The function of nuclear receptors, like EcR, is too well understood to gloss over the impact of these point mutations by saying that they either prevent hormone binding [for W650A] or "co-factor" binding [for F645A]. The W650A mutation is in the binding pocket and prevents 20E binding. This prevents the loss of co-repressors and the assembly of a co-activator complex. Therefore, this forms serves as a constitutive repressor that is not responsive to 20E. The F645A substitution, by contrast is in the helix that changes conformation after 20E binding and is one of the residues involved in co-activator assembly. EcR-F645A allows 20E binding and the conformation changes that lose the co-repressors but it cannot support the co-activation. Therefore, it ca show hormone-dependent derepression but not activation.

5. Paragraph 299: This paragraph argues that expression of EcR[-A] in the L2 suppresses subdued expression and they conclude that expression of EcR-A in the L3 will enhance subdued expression because the EcR-A w650A suppresses subdued expression in the latter. They really need to show the effects of expression of the wild-type EcR-A receptor in the L3 as well as W650A. This information is important because as noted above, ectopic expression of EcR-A could act to suppress expression although likely not to the level seen for the W650A dominant negative.

6. The interpretation of the W650A data in the L3 is confusing because its expression in the third suppresses both TrpA1 and subdued [Figure 4C] but has minimal effect on Ca^2+^ responses to heat [Figure 3] and has no effect on behavioral responses to 46 C [Figure 2D]. I was hoping that the Discussion would pull things together into a consistent story, but the Discussion tended to invoke a number of ad hoc hypotheses to account for pieces of data which do not seem to fit.

*Reviewer #3:*

The manuscript by Jaszczak et al. examines how steroid hormone signaling in *Drosophila* activates in sensory switch in C4da neurons that promotes a thermal nociceptive escape response behavior. The authors show that a greater percent of third instar larvae are able to generate the thermal nociceptive escape response than second instar larvae. They also show calcium responses in the soma of C4da neurons are larger in third instar larvae in response to high temperatures in comparison to second instar larvae. In contrast, they see the opposite with UV light. Both the behavioral response and the calcium response can be increased in second instar larvae by increasing ecdysone signaling. The authors also find that subdued expression is suppressed by increasing ecdysone signaling and link subdued to this developmental transition. Overall, the findings in the manuscript are interesting, but I think the authors need to address several points to strengthen their conclusions.

1. Some explanation as to why the soma is being imaged instead of dendrites or axons is needed. I assume the size of dendrites and axons make it too difficult to image in intact larvae.

2. The authors mention that other groups have shown that EcR regulates dendritic growth. Could the role of EcR in this sensory switch be due to changes in dendritic growth instead of changes in calcium levels. This possibility should at least be discussed.

3. The authors use a dominant-negative approach (EcR-W650A) to reduce ecdysone signaling in third instar larvae. Dominant-negative approaches are difficult to interpret. The authors should use an EcR RNAi to knockdown ECR in C4da neurons. This should be done for both the thermal nociceptive escape responses and for the calcium imaging experiments.

4. Why does the approach dominant-negative (EcR-W650A) reduce subdued expression in third instar larvae? I would expect that reduced ecdysone signaling would enhance subdued expression.

5. TrpA1 levels in second instar larvae are not affected by increased ecdysone signaling (overexpression of EcR) in second instar larvae, but they are reduced by expression of a dominant-negative (EcR-W650A). Again, this experiment should be repeated with an EcR RNAi for third instar larvae.

6. If TrpA1 levels are regulated by ecdysone, does reducing TrpA1 in C4da neurons in third instar larvae reduce both the nociceptive response and calcium levels.

7. Subdued RNAi increases the nociceptive response and C4da calcium levels in second instar larvae. Does overexpression of Subdued reduce the nociceptive response and C4da calcium levels in third instar larvae? This experiment would nicely show that Subdued levels are required for this developmental transition.

8. UAS controls are missing in figures 3 and 4.

[Editors’ note: further revisions were suggested prior to acceptance, as described below.]

Thank you for resubmitting your work entitled “Steroid hormone signaling activates thermal nociception during *Drosophila* peripheral nervous system development” for further consideration by *eLife*. Your revised article has been evaluated by K VijayRaghavan (Senior Editor) and a Reviewing Editor.

The manuscript has been improved but there are some remaining issues that need to be addressed, as outlined below:

Essential revisions:

All revisions requested can be addressed by appropriate changes to the text.

In a revised discussion, please acknowledge and mention outstanding issues that need to be addressed by future physiological or calcium imaging experiments. Also, please make appropriate revisions in response to all of the itemised requests from Reviewer 3.

*Reviewer #1:*

The authors have improved the manuscript with their additional data and revisions. They have addressed most of my comments. The one issue that remains is the removal of the data for calcium levels at different developmental stages in response to temperature. The data in the manuscript shows calcium responses during periods of UV treatment are reduced in third instar larvae compared to second instar larvae. Subdued levels are reduced in third instar larvae and knockdown of subdued reduces calcium responses in second instar larvae. The question that remains is what happens to calcium levels in response to temperature at these developmental stages. I think these data are important in understanding how this developmental switch and subdued affects thermal nociception.

*Reviewer #2:*

This revision by the Jan group has addressed many of the issues that were raised in the first round of review. Specifically, staging of larvae has been improved especially with respect to the analysis of EcR immunostaining. Similarly, the text has been revised to address nuances related to EcR isoforms and ecdysone action across the L2-L3 larval stages. Finally, knockdown controls have been provided for RNAi transgenes, bolstering the conclusions. The removal of the calcium-imaging data for technical reasons is a little bit of a shame but the basic conclusions still stand on the improved behavioral and transcriptional analysis. The finding that EcRA and subdued are regulators of the L2-L3 transition in nociceptive sensitivity will be of interest to the fly field and those researchers more generally interested in mechanisms of modulation of nociceptive sensitivity.

*Reviewer #3:*

The resubmission of this paper is greatly improved over the original. I only have a few problems with the current version.

The first is with the Figures: I do not understand their philosophy for selecting data for the supplemental figures. The paper does not have many figures and the ones that they have are relatively simple. I do not see why important data are moved into supplemental figures. Figure 3 is a case in point. Part 3B presents third instar data showing that the W650A dominant negative (DN), suppresses the response to a 42C probe while the F645A-DN does not. The response latency data for this temperature are then shown in a supplemental Figure [lacking data for F645A-DN]. The supplemental Figure also shows the important data for the 46C probe showing both % response (for W650A-DN and F645A-DN) and latency (this also lacking F645A-DN data). It would be more helpful to the reader to have them combined in a single Figure. The same could be said for combining Figure 4 with Figure 4 Supp 1A-C.

In the body of Figures 3 through 5 they represent the isoforms with a period between EcR and the isoform designation (i.e. as “EcR.A” rather than “EcR-A”). They use the correct designations in the text and the figure legends (Also in Figure 4, Supp 1E-G). They should change the Figures to have a dash between EcR and its isoform designation.

Figure 2, Supp 1 A could confuse a reader into thinking that there are actually four EcR isoforms, the last being EcR-C. The last is a truncated, experimental construct. I suggest that the authors give it a name that reflects this – perhaps δ-EcR (using the Greek letter). This designation should be used throughout the text and in Figure 4A as well as Figure 2.

Figure 5: and supplement: I assume that in ppk-GAL4/EcR.A is actually ppk-GAL4/UAS-EcR.A.

The second problem is with the use of AF1 domains when one should be talking about A/B domains. In the Discussion the authors nicely come to grips with the complexity of ecdysone signaling and the diversity of its receptors. In the results, though, some of their Interim conclusions become confusing, especially with their usage of "AF1" rather than“"A/”" domain. AF1 refers to a functionality while A/B refers to the variable, N-terminal region of the receptor. If the terms are not used properly it can be confusing because EcR-A has an A/B domain but it has no AF1 function (see Hu et al., 2003). I suggest the following changes

Line 195: This sentence would be more clear as:“"the binding of ecdysone to EcR causes the recruitment of co-activators and the loss of co-repressors”"

Line 200 better to call it the“"co-activator recruitment domai”".

Line 204: F645A does block activation, but since it allows hormone binding, ligand-dependent de-repression can occur. W650A should prevent both de-repression and activation.

Line 219. Is this really correct that ecdysone“"activatio”" of EcR is required? Your data show that with the F645A-DN (which blocks ligand-dependent activation) show the appropriate shift in nociception. All of your data suggest that ecdysone is acting via de-repression

Line 233 More properly:“"difference between EcR isoforms is within their A/B domain”".

Line 235: should use Δ-EcR rather than EcR-C; same for ln 237, and 240.

Line 236: … to all isoforms and no A/B region.

Line 241-42: this is where it becomes confusing because EcR-A does not have a AF1 domain! The fact that EcR-A is effective while δ-EcR is not suggests that the A/B region of EcR-A carries a unique repressor function. Ecdysone binding is required to lose this repression (hence the differences between W650A and F645A). The paper gets around to this idea in the Discussion but the logic through the results is a bit muddy.

Line 277: better as …the EcR-A A/B domain and ligand binding are …..

Line 392 …. The unique A/B domain of the EcR-A isoform …..

The third issue that the paper focuses on the role of the isoforms so there should at least be images of EcR-A immunostaining along with EcR-common in Figure 2.

---

## [Author Response]

[Editors’ note: the authors resubmitted a revised version of the paper for consideration. What follows is the authors’ response to the first round of review.]

Reviewer #1:This study by the Jan group examines the onset/development of functional thermal (and UV) nociception in *Drosophila* larvae. Previously, other groups had established that there are differences in nociceptive responses between second and third instar larvae (Cox) and that ecdysone receptor function is required for functional nociceptor development (Ganter). This study does a more thorough job than that previous work in documenting the stage differences and identifies a molecular player (subdued) that appears to be required for stage-specific changes in sensitivity. Overall, the data are sound and convincing and the paper is very well written. A few suggestions for improvement/clarification are offered.1. Data organization. Some of the best data showing the stage-specific difference is in supplemental figure 1C- latency. This should really be in the main figure 1- probably as panel B. Figure 1C could benefit and would be more complete if second instar data were shown for 25 and 36 degrees.

We moved the latency data from supplemental figure 1C to the main figure 1 as Figure 1B. The original thermo-calcium data of Figure 1C has been removed due to technical issues that reduced our confidence in comparing calcium levels between different developmental stages at different temperatures.

2. The baseline calcium levels in second versus third instar is quite different- much higher in second instar looking at figure 1B. This data is essentially normalized/compared to same stage's baseline but this does raise the question: Does EcR function or subdued function shift the magnitude of response or shift the second instar (or third instar) baseline? It's a little hard to tell because Figure 3 (EcR) doesn't have stills of representative cells and Figure 6 (subdued, temperature) does not either. All we see here is the ratio data. Representative still data should be presented and whether the gene affects baseline calcium versus magnitude of change upon stimulus should be discussed.

Due to technical issues that reduced our confidence in comparing calcium levels between different developmental stages at different temperatures, we decided to remove the thermo-calcium data.

3. Figure 2 organization: Personal preference but the 20E data currently in supplement 2B is probably more important to the main point than the third instar data (Main figure 2D).

We moved 20E data from supplemental 2B to the main figure 3 as Figure 3A. Additionally, to better discuss the importance of the third instar EcR data, we have reformatted Figure 3 and moved the third instar dominant negative data to Figure 3B.

4. Some of the discussion of the differences between second and third instar seems a bit oversimplified, especially as it refers to the EcR expression data in Figure 2 supplement. It's not really accurate to say that second instar has low EcR and third has high. Reality (from the data shown) is that nuclear EcR is high in early second and later third and low in late second and early third. There isn't really data to address intra-stage differences in responsiveness so this should be described a bit more clearly- without the current oversimplification. Adding modifiers (late second instar, early third instar) would help.

To better examine the expression of EcR in second and third instar we have performed new IHC staining with improved developmental staging and quantification of EcR presence in the nucleus and cytoplasm. We have presented this data in a newly formatted Figure 2. We find that earlier in larval development, EcR is evenly distributed between the nucleus and cytoplasm in the second instar and at the beginning of the third instar. Nuclear localization increased 8 hrs after larval ecdysis and in wandering third instar larvae.

5. Analysis of subdued: Most helpful would be confirmation that this RNAi line is actually reducing subdued transcript levels, as one would predict. This would enhance confidence that there are no off-target effects. Also helpful would be the converse experiment to RNAi loss of function. Authors perform Subdued RNAi and see changes in second instar thermal responsiveness. Does overexpression of Subdued shift responsiveness in third instar?

We have now measured the ability of Subdued RNAi to reduce transcript levels by qRT-PCR and included the data in Figure 6 – supplemental figure 1C.

Overexpression of Subdued in C4da neurons has previously been reported by Jang et al. 2015, and the responsiveness of third instar larvae did not change. However, when Jang et al. 2015 used a subduedGal4 to drive Subdued overexpression, the number of fast responding third instar larvae increased. We have added discussion of these data to our Discussion section (Lines 437-443).

Reviewer #2:The manuscript deals with the basis of an intriguing shift in nociceptive behavior shown by *Drosophila* larvae as they move from the L2 to the L3 instar. Prior publications have shown that the shift is controlled by ecdysone and implicated the EcR-A knock-down of EcR-A levels rendered the neurons hyposensitive to noxious thermal and mechanical stimulation and also altered features of their anatomy. Reduction in EcR-B1 did not effect their responses but has anatomical ramifications. This paper extends these studies into Ca^2+^ data from specific da neurons and also reveals a gene, subdued, that may play a role in the change in responsiveness.

Work by McParland et al. 2015 has shown that Ecdysone Receptor is required for nociception in the third instar, but it has remained unknown as to (A) what mechanism of EcR activity is required for nociception in the third instar (ligand-dependent or independent) and (B) whether ecdysone and EcR are involved in the developmental change of increased nociceptive behavior from early instars to the third instar.

There are a number of substantial concerns:1. The paper focuses on the differences between the L2 and L3 instars. According to the Methods, their L2 stage is in the middle of the L2 period, but the L3 stage is a wandering L3 larva which is at least 50 hours older. There are three critical ecdysone-related events that occur in this 50 hour span: the critical weight at about 8-10 hr after L3 ecdysis, the mid-L3 transition, and the ecdysone peak that initiates wandering. If the transition occurs during the transition from L2 to L3, then the "critical weight" transition would be a good candidate and L3 larvae about 12 to 18 hr after ecdysis would be more appropriate.

To more precisely examine the expression of EcR in second and third instar we have performed new IHC staining with improved developmental staging and quantification of EcR presence in the nucleus and cytoplasm. We have presented this data in a newly formatted Figure 2. We find that earlier in larval development, EcR is evenly distributed between the nucleus and cytoplasm in the second instar and at the beginning of the third instar. Nuclear localization increased 8 hrs after L3 ecdysis and in wandering third instar larvae.

2. Another troubling methodological issue is that their behavioral tests were run at either 42 or 46 C while the Ca^2+^ measurements are done at 44 C. This becomes a concern in a situation like in Figure 2 where W650A expression is behaviorally effective at 42 C but has no effect at 46 C. We are then provided with a Ca^2+^ response at 44 C. Since this temp is half-way between the two used for the behavior tests, we have no way of assessing whether or not the Ca^2+^ response tracks the behavioral response.

We revised the manuscript to make a clear comparison of behavioral phenotypes with a 42^o^C probe as shown in the main figure: we have moved the 46^o^C data to supplement figures while expanding our explanation for preforming these experiments and our conclusions from these data. We have removed the calcium measurements after running into technical issues that reduced our confidence in comparing calcium levels between different developmental stages at different temperatures.

3. A serious interpretational issue has to do with ignoring the complexity imposed by the EcR isoforms. Through most of the paper and figures, the authors speak of “EcR” when they really should be saying “EcR-A”. This is not a trivial issue because the EcR-B isoforms have a strong hormone-independent transactivation domain at their N-terminus whereas EcR-A does not. At times when EcR-B1 is a present, the expression of wildtype EcR-A dampens the cellular response by competing with the endogenous EcR-B1 [Schubiger et al., 2003, Mech. Dev. 120:909]. Previous experiments (McPharland et al.) used RNAi to selectively knockdown different isoforms and these suggested a function for EcR-A. Ectopic expression of EcR-A alone, without EcR-B1 for comparison, can lead to wrong interpretations, especially since both isoforms are found in the da neurons (see McPharland).

We have addressed these concerns by: (1) Specifying which EcR isoform is being discussed, (2) Measuring EcR-A and EcR-B1 protein levels by using specific targeting antibodies in Figure 2, (3) Comparing the effects of overexpression of EcR-A, EcR-B1, and C on nociception in second and third instar larvae, and (4) Including an explanation of the interactions between EcR isoforms and interpretation of the overexpression experiments (Lines 232-246).

4. Another problem is with the interpretation of the EcR dominant negatives (EcR-DN). The function of nuclear receptors, like EcR, is too well understood to gloss over the impact of these point mutations by saying that they either prevent hormone binding [for W650A] or "co-factor" binding [for F645A]. The W650A mutation is in the binding pocket and prevents 20E binding. This prevents the loss of co-repressors and the assembly of a co-activator complex. Therefore, this forms serves as a constitutive repressor that is not responsive to 20E. The F645A substitution, by contrast is in the helix that changes conformation after 20E binding and is one of the residues involved in co-activator assembly. EcR-F645A allows 20E binding and the conformation changes that lose the co-repressors but it cannot support the co-activation. Therefore, it ca show hormone-dependent derepression but not activation.

We greatly appreciate the advice regarding how our writing about the activities of the EcR mutants can be improved. We have made revisions throughout the entire manuscript, including the explanation for the activity of the function of nuclear receptors (Lines 98-107) and the effects of the point mutations on their function (Lines 196-212).

5. Paragraph 299: This paragraph argues that expression of EcR[-A] in the L2 suppresses subdued expression and they conclude that expression of EcR-A in the L3 will enhance subdued expression because the EcR-A w650A suppresses subdued expression in the latter. They really need to show the effects of expression of the wild-type EcR-A receptor in the L3 as well as W650A. This information is important because as noted above, ectopic expression of EcR-A could act to suppress expression although likely not to the level seen for the W650A dominant negative.

We have tested the effect of expression of wild-type EcR-A expression on third instar nociception and found it neither reduced nor increased nociception. We have included this data in Figure 4 —figure supplement 2.

6. The interpretation of the W650A data in the L3 is confusing because its expression in the third suppresses both TrpA1 and subdued [Figure 4C] but has minimal effect on Ca^2+^ responses to heat [Figure 3] and has no effect on behavioral responses to 46 C [Figure 2D]. I was hoping that the Discussion would pull things together into a consistent story, but the Discussion tended to invoke a number of ad hoc hypotheses to account for pieces of data which do not seem to fit.

For the transcriptional data, upon further analysis of additional controls and nociceptive genes, as well as comparison to expression of EcR-W650A in second instar and third instar expression of EcR-A RNAi, we have found a broad effect of EcR-W650A on nociceptor transcription (Figure 5 —figure supplement 12) (we have moved the 46^o^C data to supplement figures.). Due to the broad effect of EcR-W650A on nociceptor transcription in second and third instar, and the limited effect of EcR-RNAi, we have narrowed our hypotheses in the discussion to the requirement of EcR-A ligand activity for maintaining appropriate expression of nociceptive genes, and the presence of EcR-A ligand-independent pathways which promote nociception in the third instar.

Reviewer #3:The manuscript by Jaszczak et al. examines how steroid hormone signaling in *Drosophila* activates in sensory switch in C4da neurons that promotes a thermal nociceptive escape response behavior. The authors show that a greater percent of third instar larvae are able to generate the thermal nociceptive escape response than second instar larvae. They also show calcium responses in the soma of C4da neurons are larger in third instar larvae in response to high temperatures in comparison to second instar larvae. In contrast, they see the opposite with UV light. Both the behavioral response and the calcium response can be increased in second instar larvae by increasing ecdysone signaling. The authors also find that subdued expression is suppressed by increasing ecdysone signaling and link subdued to this developmental transition. Overall, the findings in the manuscript are interesting, but I think the authors need to address several points to strengthen their conclusions.1. Some explanation as to why the soma is being imaged instead of dendrites or axons is needed. I assume the size of dendrites and axons make it too difficult to image in intact larvae.

Due to technical issues that reduced our confidence in comparing calcium levels in different developmental stages at different temperatures, we decided to remove the thermo-calcium data.

2. The authors mention that other groups have shown that EcR regulates dendritic growth. Could the role of EcR in this sensory switch be due to changes in dendritic growth instead of changes in calcium levels. This possibility should at least be discussed.

Measurement of dendrite architecture in second instar larvae overexpressing EcR-A has been added to the text and to Figure 4—figure supplement 1.

3. The authors use a dominant-negative approach (EcR-W650A) to reduce ecdysone signaling in third instar larvae. Dominant-negative approaches are difficult to interpret. The authors should use an EcR RNAi to knockdown ECR in C4da neurons. This should be done for both the thermal nociceptive escape responses and for the calcium imaging experiments.

By adding new data beyond what has been previously reported with EcR-RNAi (McParland et al. 2015) and comparing the effects of EcR-DN with those of EcR-RNAi, we now show nociceptor gene expression with expression of either EcR-DN or EcR-A RNAi in Figure 5 —figure supplement 2. While we did not find detectable transcriptional changes in nociceptor genes in third instar C4da neurons with EcR-A RNAi knockdown, we found broad suppression of nociceptor genes in third instar C4da neurons with EcR ligand binding mutant (W650A) expression (Figure 5 —figure supplement 2). We have incorporated this data into our discussion of the hypothesis that EcR-A ligand activity may function in a de-repressive capacity (Lines 383-407).

4. Why does the approach dominant-negative (EcR-W650A) reduce subdued expression in third instar larvae? I would expect that reduced ecdysone signaling would enhance subdued expression.

In order to further examine this effect of EcR-DN, we have compared nociceptor gene expression with EcR-DN at second and third instars (Figure 5 —figure supplement 2) and found a broad reduction of nociceptor gene expression at both stages of development. We have added to the Discussion the hypothesis that ligand dependent de-repression may be required for appropriate regulation of nociceptor gene transcription.

5. TrpA1 levels in second instar larvae are not affected by increased ecdysone signaling (overexpression of EcR) in second instar larvae, but they are reduced by expression of a dominant-negative (EcR-W650A). Again, this experiment should be repeated with an EcR RNAi for third instar larvae.

We have compared nociceptor gene expression with expression of either EcR-DN or EcR-A RNAi (Figure 5 —figure supplement 2). While we did not find detectable transcriptional changes in nociceptor genes in third instar C4da neurons with EcR-A RNAi expression, we found broad suppression of nociceptor genes in third instar C4da neurons with EcR ligand binding mutant (W650A) expression (Figure 5 —figure supplement 2).

6. If TrpA1 levels are regulated by ecdysone, does reducing TrpA1 in C4da neurons in third instar larvae reduce both the nociceptive response and calcium levels.

Upon further analysis of additional controls and nociceptive genes, as well as comparison to expression of EcR-W650A in second instar and third instar, we have found EcR-DN does not reduce TrpA1 in third instar larvae, while EcR-W650A has a broad effect on nociceptor transcription (Figure 5 —figure supplement 1-2). Reducing TrpA1 expression in third instar larvae has previously been shown to reduce nociception (Zhong et al. 2012; Terada et al. 2016; Gu et al. 2019) – we have referred to these previous studies in the revised manuscript.

7. Subdued RNAi increases the nociceptive response and C4da calcium levels in second instar larvae. Does overexpression of Subdued reduce the nociceptive response and C4da calcium levels in third instar larvae? This experiment would nicely show that Subdued levels are required for this developmental transition.

Overexpression of Subdued in C4da neurons has previously been reported by Jang et al. 2015 to yield no effect on the responsiveness of third instar larvae. However, when we used a subdued-Gal4 to drive Subdued overexpression, the number of fast responding third instar larvae increased. We have added these data to our discussion (Lines 437-443).

8. UAS controls are missing in figures 3 and 4.

Controls have been added for qRT-PCR experiments in Figure 5.

[Editors’ note: what follows is the authors’ response to the second round of review.]

Essential revisions:Reviewer #3:The resubmission of this paper is greatly improved over the original. I only have a few problems with the current version.The first is with the Figures: I do not understand their philosophy for selecting data for the supplemental figures. The paper does not have many figures and the ones that they have are relatively simple. I do not see why important data are moved into supplemental figures. Figure 3 is a case in point. Part 3B presents third instar data showing that the W650A dominant negative (DN), suppresses the response to a 42C probe while the F645A-DN does not. The response latency data for this temperature are then shown in a supplemental Figure [lacking data for F645A-DN]. The supplemental Figure also shows the important data for the 46C probe showing both % response (for W650A-DN and F645A-DN) and latency (this also lacking F645A-DN data). It would be more helpful to the reader to have them combined in a single Figure. The same could be said for combining Figure 4 with Figure 4 Supp 1A-C.

We have combined the Figure 3 and the supplement into a single figure. We have also added the latency data for the F645A genotypes. We have combined Figure 4 and Figure 4 —figure supplement 1 A-C into the primary Figure 4. We have also added the latency data for all of the EcR isoforms and the F645A genotypes and reference to data in the text (Line 213).

In the body of Figures 3 through 5 they represent the isoforms with a period between EcR and the isoform designation (i.e. as "EcR.A" rather than "EcR-A"). They use the correct designations in the text and the figure legends (Also in Figure 4, Supp 1E-G). They should change the Figures to have a dash between EcR and its isoform designation.

Figures 3, 4 , 4—figure supplement 1, 4—figure supplement 2, 5, and 5—figure supplement 1 have been corrected to change “.” to “-“ in EcR isoform labels.

Figure 2, Supp 1 A could confuse a reader into thinking that there are actually four EcR isoforms, the last being EcR-C. The last is a truncated, experimental construct. I suggest that the authors give it a name that reflects this – perhaps δ-EcR (using the Greek letter). This designation should be used throughout the text and in Figure 4A as well as Figure 2.

The label for the synthetic isoform has been changed to “EcR-ΔC” in the text and Figure 2 —figure supplement 1 and Figure 4. Additional designation as a “synthetic isoform” has also been added to the text and figure legend.

Figure 5: and supplement: I assume that in ppk-GAL4/EcR.A is actually ppk-GAL4/UAS-EcR.A.

“UAS-“ has been added to the transgene labels in figure 5 and Figure 5 —figure supplement 1.

The second problem is with the use of AF1 domains when one should be talking about A/B domains. In the Discussion the authors nicely come to grips with the complexity of ecdysone signaling and the diversity of its receptors. In the results, though, some of their interim conclusions become confusing, especially with their usage of "AF1" rather than "A/B" domain. AF1 refers to a functionality while A/B refers to the variable, N-terminal region of the receptor. If the terms are not used properly it can be confusing because EcR-A has an A/B domain but it has no AF1 function (see Hu et al., 2003). I suggest the following changesLine 195: This sentence would be more clear as: "the binding of ecdysone to EcR causes the recruitment of co-activators and the loss of co-repressors."

Sentence changed as advised.

Line 200 better to call it the "co-activator recruitment domain".

Sentence changed as advised.

Line 204: F645A does block activation, but since it allows hormone binding, ligand-dependent de-repression can occur. W650A should prevent both de-repression and activation.

Line 200:206 changed to clarify that “W650A mutation prevents ligand binding and disrupts both derepression and activation. The F645A mutation cannot mediate activation, but retains the ligand binding capacity”. Line 202:204 has been added to clarify that ”…differences between the effect of overexpression of these mutant EcR-A constructs are likely due to differences in ligand binding ability”.

Line 219. Is this really correct that ecdysone "activation" of EcR is required? Your data show that with the F645A-DN (which blocks ligand-dependent activation) show the appropriate shift in nociception. All of your data suggest that ecdysone is acting via de-repression

Line 226 changed to clarify that “ligand activity through EcR-A is required…”.

Line 233 More properly: "difference between EcR isoforms is within their A/B domains".

Sentence changed as advised.

Line 235: should use Δ-EcR rather than EcR-C; same for ln 237, and 240.

EcR-ΔC is now used throughout the text.

Line 236: … to all isoforms and no A/B region.

Wording changed as advised.

Line 241-42: this is where it becomes confusing because EcR-A does not have a AF1 domain! The fact that EcR-A is effective while δ-EcR is not suggests that the A/B region of EcR-A carries a unique repressor function. Ecdysone binding is required to lose this repression (hence the differences between W650A and F645A). The paper gets around to this idea in the Discussion but the logic through the results is a bit muddy.Line 277: better as …the EcR-A A/B domain and ligand binding are …..

Wording changed as advised.

Line 392 …. The unique A/B domain of the EcR-A isoform …..

Wording changed as advised.

The third issue that the paper focuses on the role of the isoforms so there should at least be images of EcR-A immunostaining along with EcR-common in Figure 2.

Images of EcR-A and EcR-B1 staining have been added to Figure 2.